# TEACH: Temporal Variance-Driven Curriculum for Reinforcement Learning

## Abstract

Reinforcement Learning (RL) has achieved significant success in solving single-goal tasks. However, uniform goal selection often results in sample inefficiency in multi-goal settings where agents must learn a universal goal-conditioned policy. Inspired by the adaptive and structured learning processes observed in biological systems, we propose a novel Student-Teacher learning paradigm with a Temporal Variance-Driven Curriculum to accelerate Goal-Conditioned RL. In this framework, the teacher module dynamically prioritizes goals with the highest temporal variance in the policy's confidence score, parameterized by the state-action value (Q) function. The teacher provides an adaptive and focused learning signal by targeting these high-uncertainty goals, fostering continual and efficient progress. We establish a theoretical connection between the temporal variance of Q-values and the evolution of the policy, providing insights into the method's underlying principles and convergence guarantees. Our approach is algorithm-agnostic and integrates seamlessly with existing RL frameworks. We demonstrate this through evaluation across 11 diverse robotic manipulation and maze navigation tasks. The results show consistent and significant improvements over state-of-the-art curriculum learning and goal-selection methods.

## 1 Introduction

Deep Reinforcement Learning (DRL) (Li, 2017) has been successfully applied in solving complex problems such as robotic manipulation (Han et al., 2023), flight control (Kaufmann et al., 2018), intelligent perception system (Chaudhary et al., 2023) and real-time strategy gameplay (Andersen et al., 2018). Building upon DRL's success, learning a generalized policy to solve multiple goal-oriented tasks is of huge interest. In the past, the multi-goal or multi-task (Tzannetos et al., 2023) problems have been addressed by merging Automatic Curriculum Learning (ACL) (Bengio et al., 2009; Karni et al., 1998; Portelas et al., 2020b; Akkaya et al., 2019; Team et al., 2021; Racaniere et al., 2019) with policy learning. Broadly, ACL resorts to presenting goal or task in increasing order of difficulty (Molina & Jouen, 1998; Yengera et al., 2021; Zhang et al., 2020), and has been successfully applied to Sim2Real transfer (Akkaya et al., 2019), continuously-parameterized environments (Portelas et al., 2020a), sequencing of components in multi-agent systems (Vinyals et al., 2019), and to improve DRL agent's sample efficiency and performance (Horgan et al., 2018; Flet-Berliac & Preux, 2019).

This paper focuses on ACL from the perspective of Goal Conditioned Reinforcement Learning (GCRL) (Schaul et al., 2015). Specifically, we address multi-goal sparse reward scenarios where the agent is trained to learn a universal policy to attain any goal in the goal space, and a binary reward is provided when the desired goal is achieved. In such multi-goal scenarios under ACL, learning is most effective when the agent engages with goals positioned at the skill frontier—neither too easy nor too difficult (Pinto et al., 2017; Sukhbaatar et al., 2017). Based on this observation, (Portelas et al., 2020a; Matiisen et al., 2019) use a curriculum that samples tasks with high Learning Progress (LP), while Value Disagreement Sampling (VDS) (Zhang et al., 2020) samples goals with high epistemic uncertainty. Although ACL has improved their performance, these approaches rely on noisy value estimates to design a curriculum, and these noisy estimates become inefficient in a high-dimensional goal space with sparse reward settings.

Concretely, we are motivated by theoretical analysis on ACL for linear regression models (Weinshall et al., 2018) and curriculum design for teachers via demonstration (Yengera et al., 2021). Our key insight is that relying on uniform goal sampling to update policy and value function neglects the intrinsic dynamics of learning, where certain goals require more focus due to their evolving contributions to policy improvement and value function learning. To address this, we propose a curriculum learning framework based on temporal variance in Q-values, highlighting regions where the learning dynamics are active. Although this curriculum is defined using Q-value estimates, we establish that it inherently captures the co-evolution of both the policy and value function, a connection we formalize mathematically to provide theoretical support.

Specifically, we introduce a Student-Teacher framework for GCRL that utilizes learning progress measured by temporal variance in policy confidence score to guide curriculum design. In this framework, the teacher module (goal proposer) evaluates the agent's policy and assigns a policy confidence score to each goal in the goal space. This policy confidence score, derived from state-action value (Q) estimates, corresponds to the expected return under the current policy for a given goal. The teacher leverages these confidence scores to dynamically design the curriculum. The teacher ensures that the curriculum continuously adapts to the agent's current capabilities and promotes efficient progress. To measure progress, we introduce the concept of Learning Progress (LP), which quantifies the temporal change in policy confidence scores. The teacher evaluates learning progress by monitoring the variance in these scores over time. This variance indicates the goals at the skill frontier, where the agent's policy is evolving most rapidly. This allows the teacher to focus on regions of the goal space where the agent's learning is most active, promoting continual and effective policy improvement.

The proposed algorithm has no static design heuristic Tzannetos et al. (2023); Eimer et al. (2021) and is ensemble-free (Zhang et al., 2020). We name our approach **TEACH** (**T**emporal varianc**E** driven **A**utomatic **C**urriculum teac**H**er). Further, to demonstrate the effectiveness of TEACH, we compare the proposed algorithm with Proximal Curriculum for Reinforcement Learning (ProCurl) (Tzannetos et al., 2023) and Self-Paced Context Evaluation (SPaCE) (Eimer et al., 2021) approaches, which use value estimates for curriculum learning in contextual multi-task scenarios. We adapt their strategies to a multi-goal setting. We also compare with current state-of-the-art VDS (Zhang et al., 2020) in multi-goal GCRL. Finally, We show that TEACH consistently improves performance on different challenging tasks, including robotic manipulation (Plappert et al., 2018), dexterous in-hand manipulation (Plappert et al., 2018), and Maze navigation (Zhang et al., 2020). We summarize our contributions as follows:

- Theoretical insights: We establish a formal theoretical connection between Q-values and policy evolution, demonstrating that changes in Q-values relate to policy divergence and help identify regions of significant policy evolution.

- Curriculum design: We propose a curriculum strategy based on Learning Progress (LP) that mitigates the impact of noisy value estimates, promoting efficient learning in multi-goal sparse reward scenarios.

- Convergence analysis: We provide a formal analysis of the convergence of our curriculum learning approach, ensuring stable and consistent policy improvement.

- Validation and robustness: We validate our method through extensive experiments across diverse robotic manipulation tasks, demonstrating its effectiveness and robustness across different settings.

## 2   Related Work

Curriculum learning has been recognized as a critical factor in addressing various machine learning challenges (Selfridge et al., 1985; Elman, 1993; Bengio et al., 2009; Cangelosi & Schlesinger, 2015), aiming to structure the presentation of samples during the learning process. Although the experts can manually create such curricula tailored to specific problems, the concept of Automatic Curricula Learning (Graves et al., 2017; Portelas et al., 2020a) concentrates on developing algorithms capable of autonomously arranging the sequences of learning problems to optimize agent performance.

**Curriculum strategies for DRL:** In DRL, applications of ACL are widespread as researchers often stumble to make generalist agents and search for strategies (Cobbe et al., 2020; Zhang et al., 2020; Rajeswaran et al., 2016) that can train agents beyond their initial success. Curriculum reinforcement learning techniques (Florensa et al., 2017; 2018; Portelas et al., 2020a; Klink et al., 2021; Zhang et al., 2020; Racaniere et al., 2019; Sukhbaatar et al., 2017; Portelas et al., 2020a; Jiang et al., 2021b; Li et al., 2023; Dennis et al., 2020; Jiang et al., 2021a) primarily concentrate on enhancing an agent's learning efficiency or effectiveness across a set of tasks. Curriculum Reinforcement Learning has also been successfully extended to sim-to-real transfer by adapting domain randomization (Akkaya et al., 2019). Exploiting world models for curriculum learning have also been explored by (Hu et al., 2023; Mendonca et al., 2021; Pong et al., 2019; Pitis et al., 2020). Planning Exploratory Goals (PEG) (Hu et al., 2023) is a goal-conditioned policy that optimizes directly for goals that would result in high exploratory value trajectories. However, our focus lies on ACL design for model-free GCRL.

Automatic curriculum learning for goal-conditioned RL is an approach for presenting goals to learning agents in a meaningful order (Schaul et al., 2015; Liu et al., 2022). Hindsight Experience Replay (HER) (Andrychowicz et al., 2017) is an implicit curriculum strategy that relabels unsuccessful trajectory rollout as successful. A more robust extension of it Curriculum-guided HER (CHER) (Fang et al., 2019) uses an adaptive relabelling strategy based on diversity and goal-proximity. Combined with HER, (Zhang et al., 2020) propose a Value Disagreement Sampling (VDS) that uses a goal proposer module that prioritizes goals that maximize the epistemic uncertainty of the value function. A learning-based strategy using a Generative Adversarial Network (GAN) was used by (Florensa et al., 2018), generating goals with intermediate success probability.

Learning progress-based curriculum strategies have also been studied, which sample goals more aggressively toward which agent shows the most progress. A student-teacher framework for discrete task space that samples tasks with high LP was presented by (Matiisen et al., 2019). To extend this to continuous task space (Portelas et al., 2020a) uses a Gaussian Mixture Model (GMM) on a tuple of tasks and absolute learning progress. The tasks are then sampled from a Gaussian chosen proportionally to its mean absolute learning progress. These methods use a dense reward structure and compute LP using either the nearest neighbour (Portelas et al., 2020a) or change in episodes return for a given task. This limits their direct applicability to GCRL with binary episodic returns with random initial states. SPaCE (Eimer et al., 2021) uses the values function to design a curriculum learning based on learning progress. However, their work targets contextual RL for discrete task settings. In a similar context, ProCurl (Tzannetos et al., 2023), inspired by the pedagogical concept of *Zone of Proximal Development* (Vygotsky & Cole, 1978), uses values estimates to design a curriculum strategy that sample task that is neither too easy nor too hard.

Existing curriculum RL methods often struggle with adaptively selecting goals that maximize learning progress, relying on static heuristics (Tzannetos et al., 2023; Eimer et al., 2021) or computationally expensive uncertainty (Zhang et al., 2020) measures derived from the value function. To address this, we propose a temporal variance-driven curriculum strategy that prioritizes goals based on the variability of the policy learning progress over recent time steps. By capturing the temporal divergence in the policy learning signal, our approach adaptively identifies goals at the edge of the agent's capability without relying on static heuristics or large ensembles, ensuring efficient and continual progress.

## 3  Formal Problem Setup

**Multi-Goal MDP.** We consider a multi-goal RL setting where the objective is to learn a universal policy capable of achieving any goal in a specified goal space. The environment is defined as $\mathcal{M} = (\mathcal{S}, \mathcal{A}, \mathcal{G}, \mathcal{T}, \gamma, H, \mathcal{R})$, where $\mathcal{S}$ and $\mathcal{A}$ are the state and action spaces, $\mathcal{G} \subseteq \mathcal{S}$ is the goal space, $\mathcal{T} : \mathcal{S} \times \mathcal{A} \times \mathcal{S} \to [0, 1]$ defines the transition dynamics, $\gamma \in [0, 1)$ is the discount factor, $H$ is the episode length, and $\mathcal{R} : \mathcal{S} \times \mathcal{A} \times \mathcal{G} \times \mathcal{S}' \to \mathbb{R}$ is the goal-conditioned reward function, where $\mathcal{R}(s, a, g, s') = 0$ if $\| s' - g \| < \epsilon$ and $-1$ otherwise. We model multi-goal RL as an RL problem with an extended state space $\mathcal{S} \times \mathcal{G}$, where the policy $\pi : \mathcal{S} \times \mathcal{G} \to \mathcal{A}$ selects actions conditioned on state and goal, and the Q-function $Q : \mathcal{S} \times \mathcal{G} \times \mathcal{A} \to \mathbb{R}$ represents the goal conditioned

---

**Algorithm 1** Interplay of Teacher-Student components in RL agent training

---

1: **Initialization:** Initialize RL policy parametrized by $\theta$, replay buffer $R$, goal space $\mathcal{G}$.
2: **for** each episode $e = 1, 2, ...$ **do**
3:     Teacher (goal-proposer) picks a goal $g$ from goal space $\mathcal{G}$.
4:     Student (RL agent) generates episodic rollout trajectory $\tau_e$ aiming to reach the proposed goal $g$.
5:     Update the parameterized RL agent using off-policy updates with replay buffer $R$.
6: **end for**

---

expected return. The agent optimizes the expected discounted return over goals sampled from $\mathcal{G}$ :

$$J(\theta) = \mathbb{E}_{g \sim \mathcal{G}} \left[ \mathbb{E}_{\tau \sim \pi_{\theta_t}(\cdot|s,g)} \sum_{h=0}^{H-1} \gamma^h \mathcal{R}(s_h, a_h, g, s_h') \right]. \tag{1}$$

Where $\tau = \{(s_h, a_h)\}_{h=0}^{H-1}$ is a trajectory generated by $\pi_{\theta_t}$ conditioned on $g$.

**Curriculum Learning**. Further, we extended this multi-goal RL setup to a student-teacher paradigm. The agent's performance over goal $g$, which is uniformly sampled from goal space $\mathcal{G}$ can be defined as a policy confidence score $\mathcal{C}^{\pi_{\theta_t}}(g) = Q^{\pi_{\theta_t}}(s, g, a)$. In the context of our problem, we treat the policy as the student component and the goal proposer unit as the teacher component. The student component is updated using the goal-conditioned transitions sampled from replay buffer $R$. On the other hand, the teacher component defines a curriculum over goal space to improve the student's performance. This work focuses on the design of a teacher (curriculum) that can guide the student to learn in a sample efficient manner. The student-teacher interplay happens at the start of an episode, i.e., the teacher samples a target goal from the goal space for that particular episodic rollout. We summarize this student-teacher interplay in Algorithm 1.

# 4 Temporal Variance Driven Curriculum Design

This section introduces a novel curriculum learning approach that leverages policy confidence scores to evaluate learning progress. Existing methods primarily rely on value estimate-based metrics (Zhang et al., 2020; Eimer et al., 2021; Tzannetos et al., 2023) to design curricula, which have shown promise in multi-task RL. However, value estimates are often noisy (Libardi et al., 2021; Raileanu & Fergus, 2021) because they are trained using small, random batches of data collected from rollouts generated by the agent, which is still learning. One potential way to mitigate this issue is to use Polyak averaging (Polyak & Juditsky, 1992; Damani & Pinto, 2023) to obtain smoother value functions (refer to experiment section). Nevertheless, we argue that the effect of noisy values becomes more pronounced when employing static heuristic-based strategies (Tzannetos et al., 2023; Eimer et al., 2021).

To address this, we propose leveraging the temporal evolution of the Q-function. We posit that even when value estimates are noisy, their temporal evolution can reveal a clear direction of improvement and suppress noise in curriculum learning. Further, the temporal variance in Q-values inherently captures the co-evolution of both the policy and value function, making it more informative and less susceptible to noisy value estimates. By focusing on this temporal signal, our strategy provides a more accurate representation of the agent's learning progress, ultimately enabling the generation of a more effective learning curriculum. To this end, we first establish the theoretical connection between Q-value and policy evolution. Next, we introduce the policy confidence score, a formal metric to quantify the performance of the policy, followed by curriculum design and convergence analysis.

## 4.1 Theoretical Connection between Q-value and Policy Evolution

The Q-function, $Q^{\pi_{\theta_t}}(s, g, a) = \mathbb{E}\left[ \sum_{h=0}^{H-1} \gamma^h \mathcal{R}(s_h, a_h, g, s_h') \,\Big|\, s_0 = s, a_0 = a, g, \pi_{\theta_t} \right]$, represents the expected return from a given state-action pair under policy $\pi_{\theta_t}$ for goal $g$. Changes in the policy $\pi_{\theta_t}$ directly influence the Q-values through the Bellman operator (Bellman, 1966):

$$Q^{\pi_{\theta_t}}(s,g,a) = \mathcal{R}(s,a,g,s') + \gamma \mathbb{E}_{s' \sim \mathcal{T}(\cdot|s,a)} \left[ \mathbb{E}_{a' \sim \pi_{\theta_t}(\cdot|s',g)} Q^{\pi_{\theta_t}}(s',g,a') \right]. \tag{2}$$

As the policy evolves and improves, the state-action visitation distribution shifts, causing the Q-function to update and converge toward a new fixed point. In practical RL, Q-learning serves as a proxy for policy iteration, where temporal variance in Q-values reflects underlying changes in the policy. To formalize this, we consider a soft policy update mechanism (Haarnoja et al., 2017). While our training uses a deterministic policy via DDPG (Lillicrap et al., 2015), we employ a soft policy update model to theoretically analyze the relationship between Q-value variance and policy evolution, as it provides a tractable approximation for policy divergence.

$$\pi_{\theta_t}(a \mid s,g) \propto \exp \left( \frac{Q^{\pi_{\theta_t}}(s,g,a)}{\alpha} \right), \tag{3}$$

where $\alpha > 0$ is a temperature parameter controlling exploration. The Kullback-Leibler (KL) divergence (Kullback & Leibler, 1951) between consecutive policies is:

$$\text{KL}(\pi_{\theta_{t+1}} \parallel \pi_{\theta_t}) = \mathbb{E}_{s \sim \mathcal{D}, g \sim \mathcal{G}} \left[ \sum_a \pi_{\theta_{t+1}}(a \mid s,g) \log \frac{\pi_{\theta_{t+1}}(a \mid s,g)}{\pi_{\theta_t}(a \mid s,g)} \right]. \tag{4}$$

Expanding this using Q-values (see Appendix A for details) yields:

$$\text{KL}(\pi_{\theta_{t+1}} \parallel \pi_{\theta_t}) \approx \frac{1}{2\alpha^2} \mathbb{E}_{s \sim \mathcal{D}, g \sim \mathcal{G}} \left[ \text{Var}_{a \sim \pi_{\theta_t}(\cdot|s,g)}(\Delta Q^{\pi_{\theta_t}}(s,g,a)) \right], \tag{5}$$

where $\Delta Q^{\pi_{\theta_t}}(s,g,a) = Q^{\pi_{\theta_{t+1}}}(s,g,a) - Q^{\pi_{\theta_t}}(s,g,a)$. This approximation shows that the KL divergence between successive policies is proportional to the variance of Q-value changes, indicating that high variance in Q-values signals significant policy evolution. By tracking this variance, we gain a dual perspective on value and policy dynamics, informing our curriculum design by focusing on goals where learning is most active.

In the next subsection, we introduce the policy confidence score, a metric derived from Q-values that quantifies policy performance and guides curriculum updates.

### 4.2 Policy Confidence Score

In goal-conditioned reinforcement learning (RL), an agent learns a policy to maximize the expected sum of future rewards across a diverse set of goals sampled from a goal space $\mathcal{G}$. The effectiveness of the policy for any given goal can be intuitively assessed by the expected future reward it achieves, laying the groundwork for a metric we term the *policy confidence score*. In this subsection, we formalize this concept, ensuring it integrates seamlessly with our goal-conditioned RL framework.

In the RL framework, the agent seeks to optimize a parameterized policy $\pi_{\theta_t}(a \mid s,g)$ for the objective function formalized in equation 1. This formulation captures the agent's goal of maximizing the expected discounted return, averaged over all goals in $\mathcal{G}$. To quantify the policy's performance for a specific goal $g$, we introduce the goal-conditioned state-action value function or Q-function, as formalized in the previous section. This work adopts an actor-critic method such as Deep Deterministic Policy Gradient (DDPG) (Lillicrap et al., 2015), which directly outputs a specific action for a given state-goal pair, simplifying its application in the Q-function.

The policy parameters $\theta$ are updated to maximize $J(\theta)$ via the deterministic policy gradient:

$$\nabla_\theta J(\theta) \approx \mathbb{E}_{s \sim \mathcal{D}, g \sim \mathcal{G}} \left[ \nabla_\theta \pi_{\theta_t}(s,g) \nabla_a Q^{\pi_{\theta_t}}(s,g,a) \big|_{a=\pi_{\theta_t}(s,g)} \right], \tag{6}$$

---

**Algorithm 2** Temporal Variance-Driven Curriculum Design

---

1: **Initialization:** Initialize RL policy parametrized by $\theta$, replay buffer $R$, goal space $\mathcal{G}$, temporal window $n$ (for variance calculation), interplay frequency $\Delta$ (measured in episodes), and episode length $H$, training timesteps $T$.
2: Sample $N$ goals from goal space $\mathcal{G}$
3: **for** $t = 0, 1, 2 \ldots T$ **do**
4:   **if** $t \mod (\Delta \cdot H) = 0$ **then**
5:     Compute $\mathrm{LP}^\pi(g, t)$ over past n timesteps using equation 11 for each goal and sample target goal
6:   **else**
7:     Use the previous $\mathrm{LP}^\pi(g, t)$ to sample target goal
8:   **end if**
9:   Rollout goal-conditioned transition $(s_t, a_t, r_t, s_{t+1}, g)$ and store in $R$
10:   Update policy parameter $\theta$ using stored experience
11: **end for**

---

where $\mathcal{D}$ is the state distribution, often derived from a replay buffer in off-policy RL. This gradient reveals that policy improvement hinges on the Q-values $Q^{\pi_{\theta_t}}(s, g, \pi_{\theta_t}(s, g))$, underscoring the Q-function's role as a performance indicator during training.

Leveraging the Q-function's significance, we define the *policy confidence score* for a goal $g$ as:

$$\mathcal{C}^{\pi_{\theta_t}}(g) = \mathbb{E}_{s \sim \mathcal{D}} Q^{\pi_{\theta_t}}(s, g, \pi_{\theta_t}(s, g)), \tag{7}$$

For an actor-critic policy $\pi_{\theta_t}(s, g)$, the term $Q^{\pi_{\theta_t}}(s, g, \pi_{\theta_t}(s, g))$ represents the expected return when starting from state $s$ and pursuing goal $g$ under $\pi_{\theta_t}$. By taking the expectation over states, $\mathcal{C}^{\pi_{\theta_t}}(g)$ yields a scalar metric that reflects the policy's average performance for goal $g$, independent of specific state-action pairs. Hence, the policy confidence score $\mathcal{C}^{\pi_{\theta_t}}(g)$ provides a robust measure to evaluate the policy's capability for each goal. In the next section, we exploit this metric to devise a curriculum that prioritizes goals with high learning potential, building on the insights from equation 5, thereby improving sample efficiency and performance in goal-conditioned RL tasks.

## 4.3 Curriculum Design

To enhance learning efficiency in goal-conditioned reinforcement learning, we propose a curriculum strategy that prioritizes goals $g \in \mathcal{G}$ based on their potential to accelerate policy improvement. Specifically, we define a curriculum distribution $\mathcal{K}^\pi(g) \propto f_t(g)$, where $f_t(g)$ encodes the curriculum objective for goal $g$ at timestep $t$. This approach builds on prior work such as Value Disagreement Sampling (VDS) (Zhang et al., 2020), which uses epistemic uncertainty as $f$, and SPaCE (Eimer et al., 2021), which defines $f_t(g) = V_t^\pi(s, g) - V_{t-1}^\pi(s, g)$ to capture learning progress.

Drawing from earlier theoretical insights, equation 7, we set:

$$f_t(g) = \mathcal{C}^{\pi_{\theta_t}}(g) = \mathbb{E}_{s \sim \mathcal{D}} Q^{\pi_{\theta_t}}(s, g, \pi_{\theta_t}(s, g)), \tag{8}$$

This formulation ensures $f_t(g)$ reflects the policy's expected performance for goal $g$, averaged over a representative set of states.

**Learning Progress via Temporal Variance:** To prioritize goals with significant learning potential, we define learning progress using the temporal variance of the policy confidence score. A simple difference like $\mathcal{C}^{\pi_{\theta_t}}(g) - \mathcal{C}^{\pi_{\theta_{t-\Delta t}}}(g)$ over a fixed interval $\Delta t$ could be noisy due to unreliable Q-value estimates in reinforcement learning (Libardi et al., 2021; Raileanu & Fergus, 2021). Instead, we propose a robust metric: the temporal variance of $\mathcal{C}^{\pi_{\theta_t}}(g)$ over a window of $n$ timesteps, defined as:

$$\delta_{\mathcal{C}^\pi}(g, t) = \frac{1}{n} \sum_{k=t-n+1}^{t} \left( \mathcal{C}^{\pi_{\theta_k}}(g) - \bar{\mathcal{C}}(g, t) \right)^2, \tag{9}$$

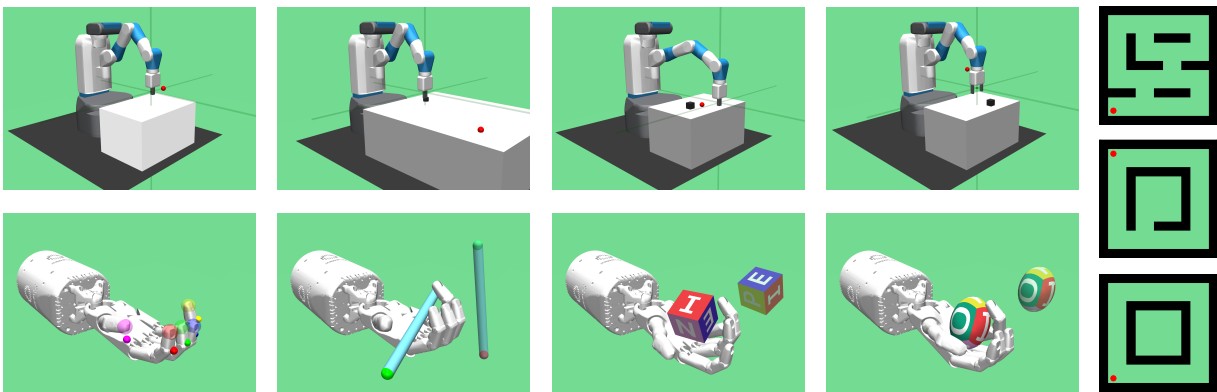

Figure 1: We evaluate the performance of TEACH on 4 FetchArm, 4 HandManipulate OpenAI Gymnasium (Plappert et al., 2018) environments, and 3 Maze navigation environments taken from (Zhang et al., 2020).

Where the mean confidence score over the window is defined as: $\bar{\mathcal{C}}(g, t) = \frac{1}{n} \sum_{k=t-n+1}^{t} \mathcal{C}^{\pi_{\theta_k}}(g)$. The learning progress for goal $g$ at timestep $t$ is then:

$$\text{LP}^{\pi}(g, t) = \frac{\delta_{\mathcal{C}^{\pi}}(g, t)}{Z_t}, \tag{10}$$

where $Z_t = \int_{\mathcal{G}} \delta_{\mathcal{C}^{\pi}}(g, t)\, dg$ normalizes the distribution over the goal space $\mathcal{G}$.

Since $\mathcal{G}$ is continuous, computing $Z_t$ directly is intractable. We address this by uniformly sampling $N$ goals $\{g_1, g_2, \ldots, g_N\} \subset \mathcal{G}$ at the start of training. For each sampled goal $g_i$, we compute $\delta_{\mathcal{C}^{\pi}}(g_i, t)$ and approximate the learning progress as:

$$\text{LP}^{\pi}(g_i, t) = \frac{\delta_{\mathcal{C}^{\pi}}(g_i, t)}{\sum_{j=1}^{N} \delta_{\mathcal{C}^{\pi}}(g_j, t)}. \tag{11}$$

The curriculum distribution becomes a discrete probability over the sampled goals and goals are sampled proportionally to these probabilities during training. This process is repeated periodically to update the curriculum based on the latest policy and Q-value estimates. The use of temporal variance as a learning progress metric leverages the correlation between significant Q-value changes and policy evolution. By prioritizing goals with high $\delta_{\mathcal{C}^{\pi}}(g, t)$, the curriculum focuses the agent on regions of the goal space where the policy is actively improving, enhancing sample efficiency. Unlike simple differences, the variance over $n$ timesteps reduces sensitivity to noisy Q-value updates, offering a reliable signal for curriculum design. This approach integrates policy and value function dynamics into a robust measure tailored to goal-conditioned RL tasks' continuous and noisy nature.

### 4.4 Convergence Analysis

Under standard Q-learning assumptions and the Hindsight Experience Replay (HER) (Andrychowicz et al., 2017) framework, TEACH ensures stable learning dynamics and convergence through a structured curriculum expansion process.

**Theorem 1** Let $\mathcal{G}$ be a finite goal set. For each goal $g \in \mathcal{G}$, Assume:

- **Q-Learning:** The standard Q-learning assumptions: The Q-network is over-parameterized with Lipschitz activations; the learning rate satisfies the Robbins-Monro conditions (Robbins & Monro, 1951)[1]; and the replay buffer is sufficiently large to approximate i.i.d. sampling.

---

[1]Polyak averaging and target networks approximately satisfy the Robbins-Monro conditions.

- **Curriculum Expansion Rule[2]:** A goal $g'$ is added to the current curriculum set $\mathcal{G}_{\text{curr}}$ when:

$$\max_{g \in \mathcal{G}_{\text{curr}}} \text{Var}_t^\pi(g) \leq \eta \quad \text{and} \quad \text{LP}^\pi(g', t) = \max_{g \in \mathcal{G}} \frac{\text{Var}_t^\pi(g)}{\overline{\text{Var}_t^\pi}(\mathcal{G})}, \tag{12}$$

  where $\text{Var}_t^\pi(g)$ is the temporal variance of the Q-values over the past $n$ timesteps, and $\overline{\text{Var}_t^\pi}(\mathcal{G}) = \frac{1}{|\mathcal{G}|} \sum_{g \in \mathcal{G}} \text{Var}_t^\pi(g)$ is the average temporal variance across all goals.

Then, with probability 1:

$$\exists T < \infty \text{ such that } \forall t > T, \quad \mathcal{G}_{\text{curr}} = \mathcal{G} \text{ and } \pi_{\theta_t} \xrightarrow{t \to \infty} \pi^*.$$

**Proof: Q-Learning Convergence:** Following the insights of (Fan et al., 2020), under the Neural Tangent Kernel (NTK) (Jacot et al., 2018) regime, the Q-function can be approximated as a near-linear model during training using standard Q-learning assumptions. In this setting, the parameter $\phi$ updates follows:

$$\phi_{t+1} = \phi_t - \alpha_t \nabla_\phi \mathcal{L}(\phi_t),$$

where $\mathcal{L}$ is the temporal difference loss and $\alpha_t$ is the learning rate at time $t$. Under NTK assumptions, the Q-values converge to a fixed point near $Q^*$ with an approximation error bounded as

$$\lim_{t \to \infty} |Q^{\pi_{\theta_t}}(s, g, a) - Q^*(s, g, a)| \leq \mathcal{O}\left( \epsilon_{\text{approx}} + \gamma \sqrt{\frac{\log T}{T}} \right),$$

where $\epsilon_{\text{approx}}$ captures the function approximation error, and $T$ denotes the number of training steps. This establishes the convergence of Q-values to the optimal Q-function up to an approximation error dependent on function representation and optimization dynamics.

**Temporal Variance Decay:** Since $Q^{\pi_{\theta_t}} \to Q^*$, the variance of Q-values over time is given by:

$$\text{Var}_t^\pi(g) = \frac{1}{n} \sum_{k=t-n}^{t-1} \left( Q^{\pi_{\theta_k}}(s, g, a) - \overline{Q^{\pi_\theta}}(g) \right)^2, \tag{13}$$

where $\overline{Q^{\pi_\theta}}(g) = \frac{1}{n} \sum_{k=t-n}^{t-1} Q^{\pi_{\theta_k}}(s, g, a)$. By the convergence of $Q^{\pi_{\theta_t}}$, it follows that $\text{Var}_t^\pi(g) \to 0$ for all $g \in \mathcal{G}$.

**Curriculum Saturation:** By the curriculum rule, a goal $g'$ is added when $\max_{g \in \mathcal{G}_{\text{curr}}} \text{Var}_t^\pi(g) \leq \eta$. Since $\text{Var}_t^\pi(g) \to 0$ for all $g$, there exists a finite timestep $T$ such that $\mathcal{G}_{\text{curr}} = \mathcal{G}$ for all $t > T$.

**Policy Convergence:** Since all goals are eventually included in $\mathcal{G}_{\text{curr}}$, the agent continues training on all $g \in \mathcal{G}$. By Q-learning convergence, the policy updates according to:

$$\pi_{\theta_{t+1}}(a|s, g) = \arg \max_a Q^{\pi_{\theta_t}}(s, g, a) \xrightarrow{t \to \infty} \arg \max_a Q^*(s, g, a) = \pi^*(a|s, g). \tag{14}$$

Thus, the policy empirically converges to the optimal policy $\pi^*$.

**Mitigation of Forgetting:** HER ensures the replay buffer contains a wide range of goals by relabeling failed trajectories with achieved goals, stabilizing learning, and mitigating catastrophic forgetting. Since transitions are uniformly sampled from the replay buffer, the training distribution approximates

---

[2]In the following theorem, we introduce a theoretical curriculum expansion rule to prove convergence, ensuring that all goals in $\mathcal{G}$ are eventually trained. This rule posits that a new goal is added to a curriculum set $\mathcal{G}_{\text{curr}}$ when the current goals are mastered (i.e., $\max_{g \in \mathcal{G}_{\text{curr}}} \text{Var}_t^\pi(g) \leq \eta$). In practice, TEACH does not maintain an explicit $\mathcal{G}_{\text{curr}}$; instead, it dynamically samples goals from $\mathcal{G}$ with probabilities proportional to $\text{LP}^\pi(g, t)$ every $\Delta$ episodes (Algorithm 2). This sampling implicitly expands the curriculum by shifting focus to goals with high learning potential as others stabilize, achieving the same coverage as the theoretical rule.

Table 1: Complexity of enviornments

| Environment | Reward | State | Context | Action | Goal space size | Episode length |
|---|---|---|---|---|---|---|
| FetchReach-v2 | binary | $\mathbb{R}^{10}$ | $\mathbb{R}^3$ | $\mathbb{R}^3$ | $1e^3$ | 50 |
| FetchPickAndPlace-v2 | binary | $\mathbb{R}^{25}$ | $\mathbb{R}^3$ | $\mathbb{R}^4$ | $1e^3$ | 50 |
| FetchSlide-v2 | binary | $\mathbb{R}^{25}$ | $\mathbb{R}^3$ | $\mathbb{R}^3$ | $1e^3$ | 50 |
| FetchPush-v2 | binary | $\mathbb{R}^{25}$ | $\mathbb{R}^3$ | $\mathbb{R}^3$ | $1e^3$ | 50 |
| HandManipulateBlock-v1 | binary | $\mathbb{R}^{61}$ | $\mathbb{R}^7$ | $\mathbb{R}^{20}$ | $1e^3$ | 100 |
| HandManipulatePen-v1 | binary | $\mathbb{R}^{61}$ | $\mathbb{R}^7$ | $\mathbb{R}^{20}$ | $1e^3$ | 100 |
| HandManipulateEgg-v1 | binary | $\mathbb{R}^{61}$ | $\mathbb{R}^7$ | $\mathbb{R}^{20}$ | $1e^3$ | 100 |
| HandReach-v1 | binary | $\mathbb{R}^{63}$ | $\mathbb{R}^{15}$ | $\mathbb{R}^{20}$ | $1e^3$ | 50 |
| MazeA-v0 | binary | $\mathbb{R}^2$ | $\mathbb{R}^2$ | $\mathbb{R}^2$ | $1e^3$ | 50 |
| MazeB-v0 | binary | $\mathbb{R}^2$ | $\mathbb{R}^2$ | $\mathbb{R}^2$ | $1e^3$ | 50 |
| MazeC-v0 | binary | $\mathbb{R}^2$ | $\mathbb{R}^2$ | $\mathbb{R}^2$ | $1e^3$ | 50 |

a stationary process, further enhancing learning stability. This aligns with empirical findings in prior work (Andrychowicz et al., 2017; Zhang et al., 2020), where HER-based training stabilizes policy updates across multiple goals. Hence, HER and uniform replay sampling empirically ensure stable convergence across all goals.
Hence, TEACH's efficiency stems from its adaptive sampling of goals with high $\text{LP}^\pi(g, t)$, which correlates with $\text{Var}_t^\pi(g)$. As the agent masters a goal (low $\text{Var}_t^\pi(g)$), the teacher shifts focus to new high-variance goals, implicitly expanding without static heuristics, unlike VDS or ProCuRL. This dynamic prioritization and HER's relabeling ensure broad goal coverage with fewer samples.

Finally, we summarize the proposed method in Algorithm 2. Our approach employs the Deep Deterministic Policy Gradient (DDPG) (Lillicrap et al., 2015) algorithm as the base RL policy. To enhance the efficiency of universal policy learning, we incorporate the Hindsight Experience Replay (HER) (Andrychowicz et al., 2017) strategy into the replay buffer manipulation. HER allows the off-policy RL algorithm to learn more effectively by reinterpreting unsuccessful trajectories as successful.

A notable advantage of our method is that the teacher component, responsible for curriculum learning, does not introduce any new hyperparameters except the temporal window ($n$), which captures the evolution of the policy confidence score over past $n$ time steps. This design choice ensures that the training routine remains consistent with the base RL policy, simplifying the implementation and reducing the potential for hyperparameter tuning complexity. As a result, our method maintains the stability and reliability of the base RL algorithm while enhancing its learning efficiency through strategic replay buffer manipulation and curriculum learning.

## 5 Experiments

We test our method on 11 multi-goal binary reward tasks. The complexity details of the task are shown in Table 1, context refers to the goal's dimension, and goal space size ($N$) refers to the number of goals sampled from continuous goal space to make the problem tractable. While our main focus relies on the robotics task environment (Plappert et al., 2018), we also include three Maze navigation tasks (Zhang et al., 2020) to evaluate performance in low contextual problems. Refer to Appendix B for task definitions.

### 5.1 Baselines

To establish the contribution and effectiveness of the proposed approach, we compare our method with the following baselines. The VDS and ProCurl are currently state-of-the-art in multi-goal and multi-task

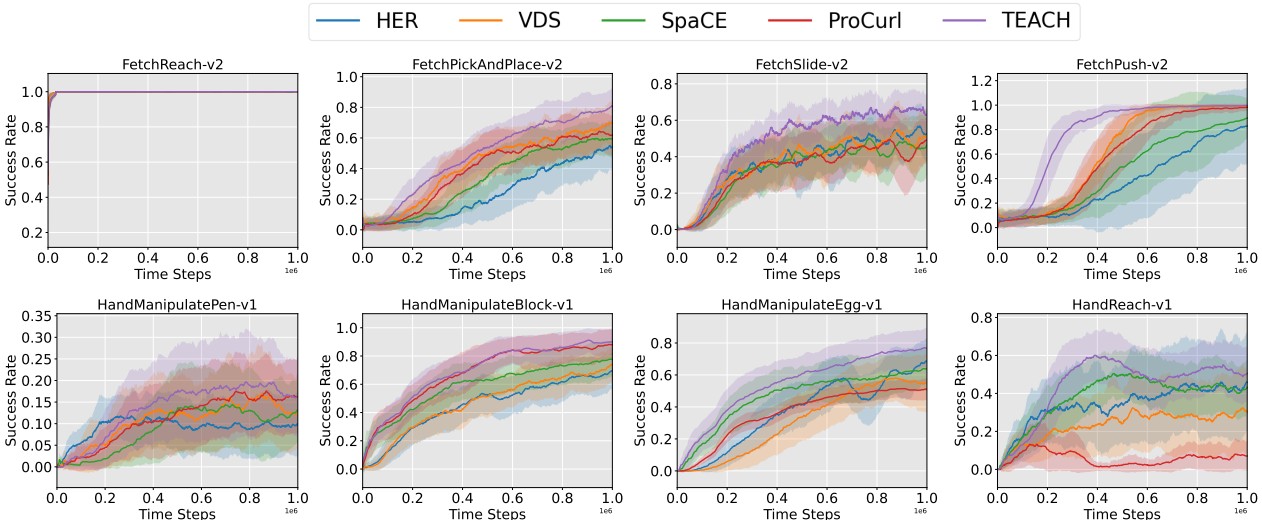

Figure 2: Results show the performance across 8 robotic manipulation tasks (refer supplementary for complete results). The plots show the success rate along the y-axis evaluated through current policy. The reported results are mean across 5 seeds with shaded regions highlighting standard deviation.

settings, respectively. All the baselines use value estimates to design ACL, which makes them a strong choice for baselines to highlight the advantage of our approach.

**HER-IID** (Andrychowicz et al., 2017) In HER-IID, the RL agent uses a hindsight experience reply buffer which independently and identically samples goals from task space. We use the official code-based implementation to reproduce the results.

**VDS** (Zhang et al., 2020) Value Disagreement Sampling samples goals from the goal space based on value disagreement. Their strategy prioritizes goals that maximize the epistemic uncertainty of the Q-function of the policy. We use the official implementation to reproduce the results.

**SPaCE** (Eimer et al., 2021) Self-Paced Context Evaluation provides a curriculum learning explicitly using agent's performance as an ordering criterion. They use the agent's state value predictions to generate curriculum. Which uses the temporal difference to design a curriculum that heuristically selects a new goal if it observes significant learning in the value function. We extended their strategy to our setting using DDPG+HER (Andrychowicz et al., 2017) implementation.

**ProCuRL** (Tzannetos et al., 2023) Proximal Curriculum for Reinforcement Learning Agents is inspired by the pedagogical concept of the zone of proximal development. They capture the idea of proximal using state value estimates w.r.t. the learner's current policy. Their strategy essentially serves as a geometric mean of value function over goals and sample goal with the highest geometric mean. We extended their strategy to our setting using DDPG+HER (Andrychowicz et al., 2017) implementation.

## 5.2 Implementation Details

Our curriculum design comprises two primary components: the teacher (goal proposer) and the student (RL policy). The teacher is a non-learning module that evaluates the student's capabilities (the RL policy) to propose goals that promote efficient learning. The RL policy, parameterized by $\theta$, is deterministic by design, and we add noise to its actions to enable better exploration. We combine DDPG with HER. Incorporating the HER replay buffer relabeling strategy enhances learning efficiency.

At the beginning of each episode, the student queries the teacher for a target goal. The teacher evaluates the change in the student's learning progress for each goal within the goal space, which consists of $N$ goals. The teacher selects and assigns the goal with maximum temporal uncertainty in the student's confidence

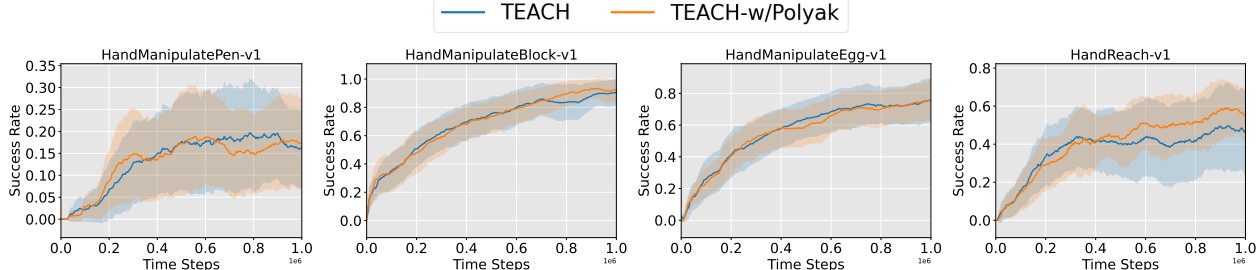

Figure 3: Results show the effect of smooth target confidence score for temporal window size $n = 3$. The results are across 5 seeds where the shaded region represents the standard deviation (refer to Figure 7 for complete results). We observe that the smooth target confidence score compared to the standard confidence score led to similar performances. This validates the hypothesis that the proposed method measure is robust to noisy value estimates.

score as the target goal. The RL policy then focuses on achieving this assigned goal. Transitions generated through agent-environment interactions are stored in the replay buffer and subsequently used to update the policy. The network architecture is MLP with 2 layers for Fetch and Maze navigation tasks and 3 layers for the HandManipulation task, respectively, for both the actor and critic networks. The Q network is trained using a learning rate of 0.001 with a batch size of 1024; refer to supplementary for hyperparameter details. All robotics tasks are trained for 1M time steps and maze navigation tasks for 400K time steps, respectively. The agent's performance is evaluated by randomly sampling goals from the goal space. All the reported results highlight the agent's success rate in achieving those randomly sampled goals. Specifically, the success rate is calculated as the average success rate in goal accomplishment, which averaged over 20 episodes (each with a randomly sampled goal) at each evaluation step.

### 5.3 Improvement Through TEACH

The performance of all methods on robotics manipulation tasks is shown in Figure 1. These results demonstrate that our proposed approach, which inherently combines dual exploitation of the information from the current policy and state-action value estimates through temporal evolution of Q-value, achieves superior sample efficiency. The performance improvements are evident across low-context tasks, such as *Maze navigation*, and high-context tasks, such as *HandManipulation*. The reported performance of TEACH is computed using a temporal window of $n = 10$, i.e., the curriculum is designed based on the temporal divergence of the Q-function over the past 10 time steps. However, we highlight in Section 5.5 that the performance of TEACH is mostly invariant with the size of the temporal history of the value function.

Our approach consistently yields better policies while maintaining high sample efficiency, as evidenced by extensive comparisons across tasks and baselines. The improvements stem from addressing two primary challenges: (1) the noisy and often inaccurate nature of value estimates and (2) the additional noise introduced by the HER relabeling strategy, which adds fictitious and biased data into the replay buffer used for training the value function.

We mitigate these challenges by designing a curriculum that better aligns with the current policy's capabilities and is less prone to the detrimental effects of noisy value estimates. By leveraging dual exploitation—drawing information from both the current policy and state-action value estimates using temporal divergence to measure learning progress, our strategy reduces the impact of noise and enhances the robustness and effectiveness of policy learning.

### 5.4 Effect of Smooth Target Confidence Score on Performance

The issue of noisy value estimates and bias can be addressed by Polyak averaging (Polyak & Juditsky, 1992) to obtain a smooth value function. The smoothed values function is often called the 'target' values function.

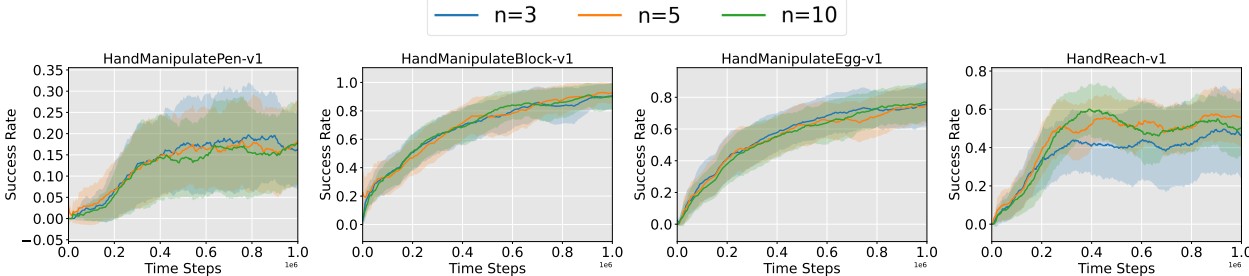

Figure 4: Ablation analysis of the effect of temporal window size $n$. The results are across 5 seeds where the shaded region represents the standard deviation (refer to Figure 8 for complete results). We observe that the presented approach is insensitive to temporal window size.

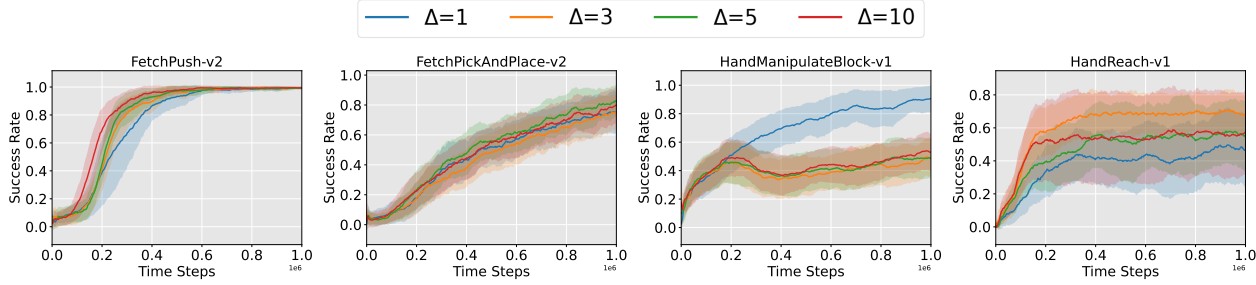

Figure 5: Ablation analysis of the effect of student-teacher interplay frequency $\Delta$. The results are across 5 seeds where the shaded region represents the standard deviation. We observe that the performance can be improved using different interplay frequencies for different tasks.

The target function parameters are computed using equation 15, where $\theta'$ are parameters of the target values function and $\theta$ are parameters of the value function.

$$\theta' = \alpha\theta' + (1 - \alpha)\theta \tag{15}$$

We conduct an ablation study incorporating a smooth state-action value function to evaluate the impact of a smooth confidence score. The results, presented in Figure 3, indicate that a smooth target state-action value function does not improve performance compared to the noisy state-action values function used in the proposed algorithm. The smooth success score provides better results in *FetchPickAndPlace* task but performs poorly for *FetchPush* task while performing similarly across the remaining tasks. This underscores the ability of the TEACH to be robust to noisy value estimates, which further validates the arguments made in the section 4. The impact of the noisy value estimates can be reduced further to some extent by using a larger temporal window size.

## 5.5 Effect of Size of Temporal Window

In this analysis, we investigate the effect of the temporal window size $n$ on our proposed automatic curriculum learning method, TEACH. As shown in Figure 4, the proposed approach demonstrates a high degree of invariance to the size of the temporal window $n$, which captures the length of past time steps used for computing temporal divergence of policy confidence score. This invariance highlights the robustness of TEACH in adapting to different temporal contexts, making it suitable for diverse task settings. However, while the overall performance remains consistent across different values of $n$, we observe that $n = 10$ produces slightly better results in some tasks. This may be because $n = 10$ strikes an optimal balance between capturing sufficient historical information and avoiding sensitivity to noise compared to smaller temporal windows. Therefore, we report results using $n = 10$ in Figure 1 for consistency and reproducibility.

### 5.6 Effect of Teacher-Student Interplay Frequency

In this ablation study, we investigate the effect of the teacher-student interplay frequency ($\Delta$) on learning performance. The interplay frequency controls the temporal interval (number of episodes), after which changes in the policy confidence score are reevaluated. As illustrated in Figure 5, a less aggressive interplay frequency can lead to faster learning. These results suggest that a continuous curriculum generation strategy may induce forgetting behaviors, as agents are exposed to diverse goals at varying time steps. Conversely, oversampling specific goals allows agents to focus on a consistent set of objectives, thereby facilitating improved learning. However, our findings reveal that achieving more stable and consistent performance requires an adaptive interplay approach, yielding superior results to fixed strategies, which may require extensive fine-tuning. While this work employs a fixed teacher-student interplay frequency ($\Delta = 1$) to underscore the benefits of a temporal divergence-driven curriculum strategy. Nevertheless, we emphasize that the design of adaptive curriculum strategies that dynamically adjust the interplay between the teacher and the student to optimize learning performance can result in superior performance. Notably, prior works have not addressed the importance of adaptability in curriculum generation, and we believe that future research should explore the design of adaptive curriculum strategies that dynamically adjust the interplay between the teacher and the student to optimize learning performance.

## 6   Conclusion

In this work, we introduced a novel curriculum strategy for goal-conditioned reinforcement learning that leverages a temporal divergence-based approach to address the challenges posed by noisy value estimates in curriculum design. We first demonstrated theoretically how temporal divergence serves as an upper bound on policy divergence, providing a more reliable mechanism for evaluating both policy performance and the value function. Building on this foundation, we provided a convergence analysis. Subsequently, we conducted extensive experiments across 11 robotics and navigation environments with binary rewards, showcasing the practical effectiveness of our strategy. In these experiments, we highlighted the robustness of our algorithm, demonstrating its insensitivity to smooth target success score estimates and its ability to mitigate the effects of noisy value estimates. Finally, we demonstrated that an adaptive curriculum learning strategy, built upon our temporal divergence-based framework, can outperform fixed strategies, yielding improved and more stable results.

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

# A    Correlation Between Q-Value and Policy Divergence

We begin by considering a soft policy update mechanism (Haarnoja et al., 2017), where the policy update rule can be defined as:

$$\pi_{\theta_t}(a|s) \propto \exp\left(\frac{Q^{\pi_{\theta_t}}(s,g,a)}{\alpha}\right), \tag{16}$$

where $\alpha > 0$ is a temperature parameter that governs the trade-off between exploration and exploitation. The partition function $Z_t(s) = \sum_a \exp(Q^{\pi_{\theta_t}}(s,g,a)/\alpha)$ normalizes the policy distribution $\pi_{\theta_t}(a|s,g)$.

To understand the changes in the policy across consecutive updates, we analyze the Kullback-Leibler (KL) divergence (Kullback & Leibler, 1951) between the two policies:

$$\mathrm{KL}(\pi_{\theta_{t+1}} \parallel \pi_{\theta_t}) = \mathbb{E}_{s\sim\mathcal{D},g\sim\mathcal{G}}\left[\sum_a \pi_{\theta_{t+1}}(a|s,g)\log\frac{\pi_{\theta_{t+1}}(a|s,g)}{\pi_{\theta_t}(a|s,g)}\right]. \tag{17}$$

By expressing the policy in the softmax form $\pi_{\theta_t}(a|s,g) = \frac{\exp(Q^{\pi_{\theta_t}}(s,g,a)/\alpha)}{Z_t(s)}$, the KL divergence expands as:

$$\mathrm{KL}(\pi_{\theta_{t+1}} \parallel \pi_{\theta_t}) = \mathbb{E}_{s\sim\mathcal{D},g\sim\mathcal{G}}\left[\sum_a \pi_{\theta_{t+1}}(a|s,g)\left(\frac{Q^{\pi_{\theta_{t+1}}}(s,g,a) - Q^{\pi_{\theta_t}}(s,g,a)}{\alpha}\right) - \log\frac{Z_{t+1}(s)}{Z_t(s)}\right]. \tag{18}$$

$$\mathrm{KL}(\pi_{\theta_{t+1}} \parallel \pi_{\theta_t}) = \mathbb{E}_{s\sim\mathcal{D},g\sim\mathcal{G}}\left[\frac{1}{\alpha}\mathbb{E}_{a\sim\pi_{\theta_{t+1}}}[\Delta Q^{\pi_{\theta_t}}(s,g,a)] - \log\frac{Z_{t+1}(s)}{Z_t(s)}\right] \tag{19}$$

Now, recall that the partition functions are given by:

$$Z_t(s) = \sum_a \exp(Q^{\pi_{\theta_t}}(s,g,a)/\alpha), \quad Z_{t+1}(s) = \sum_a \exp(Q^{\pi_{\theta_{t+1}}}(s,g,a)/\alpha), \tag{20}$$

$$\frac{Z_{t+1}(s)}{Z_t(s)} = \frac{\sum_a \exp\left(\frac{Q^{\pi_{\theta_{t+1}}}(s,g,a)}{\alpha}\right)}{Z_t(s)} = \sum_a \frac{\exp\left(\frac{Q^{\pi_{\theta_t}}(s,g,a) + \Delta Q^{\pi_{\theta_t}}(s,g,a)}{\alpha}\right)}{Z_t(s)}. \tag{21}$$

where $\Delta Q^{\pi_{\theta_t}}(s,g,a) = Q^{\pi_{\theta_{t+1}}}(s,g,a) - Q^{\pi_{\theta_t}}(s,g,a)$.

Since the policy is defined as:

$$\pi_{\theta_t}(a \mid s,g) = \frac{\exp(Q^{\pi_{\theta_t}}(s,g,a)/\alpha)}{Z_t(s)}, \tag{22}$$

We can rewrite the ratio as:

$$\frac{Z_{t+1}(s)}{Z_t(s)} = \sum_a \pi_{\theta_t}(a \mid s,g)\exp\left(\frac{\Delta Q^{\pi_{\theta_t}}(s,g,a)}{\alpha}\right) = \mathbb{E}_{a\sim\pi_{\theta_t}}\left[e^{\Delta Q^{\pi_{\theta_t}}(s,g,a)/\alpha}\right]. \tag{23}$$

$$\frac{Z_{t+1}(s)}{Z_t(s)} = \mathbb{E}_{a\sim\pi_{\theta_t}}\left[e^{\frac{\Delta Q^{\pi_{\theta_t}}(s,g,a)}{\alpha}}\right], \tag{24}$$

To simplify further, we expand the ratio of partition functions for small policy updates using a Taylor series expansion around $\Delta Q^{\pi_{\theta_t}}/\alpha \approx 0$ assuming small policy updates::

$$\frac{Z_{t+1}(s)}{Z_t(s)} = \mathbb{E}_{a\sim\pi_{\theta_t}}\left[e^{\frac{\Delta Q^{\pi_{\theta_t}}(s,g,a)}{\alpha}}\right] \approx 1 + \frac{1}{\alpha}\mathbb{E}_{a\sim\pi_{\theta_t}}[\Delta Q^{\pi_{\theta_t}}(s,g,a)] + \frac{1}{2\alpha^2}\mathbb{E}_{a\sim\pi_{\theta_t}}[(\Delta Q^{\pi_{\theta_t}}(s,g,a))^2]. \tag{25}$$

Next, taking the logarithm and using the approximation $\log(1+x) \approx x - \frac{x^2}{2}$ for small $x$, we obtain:

$$\log \frac{Z_{t+1}(s)}{Z_t(s)} \approx \underbrace{\frac{\mathbb{E}_{a \sim \pi_{\theta_t}}[\Delta Q^{\pi_{\theta_t}}(s,g,a)]}{\alpha} + \frac{\mathbb{E}_{a \sim \pi_{\theta_t}}[(\Delta Q^{\pi_{\theta_t}}(s,g,a))^2]}{2\alpha^2}}_{\text{First-order term}}$$

$$\underbrace{-\frac{1}{2}\left(\frac{\mathbb{E}_{a \sim \pi_{\theta_t}}[\Delta Q^{\pi_{\theta_t}}(s,g,a)]}{\alpha} + \frac{\mathbb{E}_{a \sim \pi_{\theta_t}}[(\Delta Q^{\pi_{\theta_t}}(s,g,a))^2]}{2\alpha^2}\right)^2}_{\text{Second-order term}}. \tag{26}$$

Expanding second-order term,

$$\left(\frac{\mathbb{E}_{a \sim \pi_{\theta_t}}[\Delta Q^{\pi_{\theta_t}}(s,g,a)]}{\alpha} + \frac{\mathbb{E}_{a \sim \pi_{\theta_t}}[(\Delta Q^{\pi_{\theta_t}}(s,g,a))^2]}{2\alpha^2}\right)^2 = \frac{(\mathbb{E}_{a \sim \pi_{\theta_t}}[\Delta Q^{\pi_{\theta_t}}(s,g,a)])^2}{\alpha^2} + \frac{(\mathbb{E}_{a \sim \pi_{\theta_t}}[(\Delta Q^{\pi_{\theta_t}}(s,g,a))^2])^2}{4\alpha^4}$$

$$+ \frac{\mathbb{E}_{a \sim \pi_{\theta_t}}[\Delta Q^{\pi_{\theta_t}}(s,g,a)] \cdot \mathbb{E}_{a \sim \pi_{\theta_t}}[(\Delta Q^{\pi_{\theta_t}}(s,g,a))^2]}{\alpha^3}. \tag{27}$$

For small $\Delta Q^{\pi_{\theta_t}}(s,g,a)$, higher-order terms like $\frac{\mathbb{E}_{a \sim \pi_{\theta_t}}[\Delta Q^{\pi_{\theta_t}}(s,g,a)] \cdot \mathbb{E}_{a \sim \pi_{\theta_t}}[(\Delta Q^{\pi_{\theta_t}}(s,g,a))^2]}{\alpha^3}$ and $\frac{(\mathbb{E}_{a \sim \pi_{\theta_t}}[(\Delta Q^{\pi_{\theta_t}}(s,g,a))^2])^2}{4\alpha^4}$ can be neglected. Thus, the second-order term simplifies to $\frac{(\mathbb{E}_{a \sim \pi_{\theta_t}}[\Delta Q^{\pi_{\theta_t}}(s,g,a)])^2}{2\alpha^2}$.

$$\log \frac{Z_{t+1}(s)}{Z_t(s)} \approx \frac{\mathbb{E}_{a \sim \pi_{\theta_t}}[\Delta Q^{\pi_{\theta_t}}(s,g,a)]}{\alpha} + \frac{\mathbb{E}_{a \sim \pi_{\theta_t}}[(\Delta Q^{\pi_{\theta_t}}(s,g,a))^2]}{2\alpha^2} - \frac{(\mathbb{E}_{a \sim \pi_{\theta_t}}[\Delta Q^{\pi_{\theta_t}}(s,g,a)])^2}{2\alpha^2}. \tag{28}$$

Factoring out $\frac{1}{2\alpha^2}$:

$$\log \frac{Z_{t+1}(s)}{Z_t(s)} \approx \frac{\mathbb{E}_{a \sim \pi_{\theta_t}}[\Delta Q^{\pi_{\theta_t}}(s,g,a)]}{\alpha} + \frac{1}{2\alpha^2}\left(\mathbb{E}_{a \sim \pi_{\theta_t}}[(\Delta Q^{\pi_{\theta_t}}(s,g,a))^2] - (\mathbb{E}_{a \sim \pi_{\theta_t}}[\Delta Q^{\pi_{\theta_t}}(s,g,a)])^2\right). \tag{29}$$

$$\log \frac{Z_{t+1}(s)}{Z_t(s)} \approx \frac{1}{\alpha}\mathbb{E}_{a \sim \pi_{\theta_t}}[\Delta Q^{\pi_{\theta_t}}(s,g,a)] + \frac{1}{2\alpha^2}\mathrm{Var}_{a \sim \pi_{\theta_t}}(\Delta Q^{\pi_{\theta_t}}(s,g,a)). \tag{30}$$

Finally, under the assumption of small policy updates ($\pi_{\theta_{t+1}} \approx \pi_{\theta_t}$) using first-order policy approximation:

$$\mathbb{E}_{a \sim \pi_{\theta_{t+1}}}[\Delta Q^{\pi_{\theta_t}}(s,g,a)] \approx \mathbb{E}_{a \sim \pi_{\theta_t}}[\Delta Q^{\pi_{\theta_t}}(s,g,a)] + \frac{1}{\alpha}\mathrm{Var}_{a \sim \pi_{\theta_t}}(\Delta Q^{\pi_{\theta_t}}(s,g,a)), \tag{31}$$

Hence, substituting equation 30 and equation 31 in equation 19 the KL divergence between two successive policy updates can be approximated as:

$$\mathrm{KL}(\pi_{\theta_{t+1}} \parallel \pi_{\theta_t}) \approx \mathbb{E}_{s \sim \mathcal{D}, g \sim \mathcal{G}}\left[\frac{1}{\alpha}\left(\mathbb{E}_{a \sim \pi_{\theta_t}}[\Delta Q^{\pi_{\theta_t}}(s,g,a)] + \frac{1}{\alpha}\mathrm{Var}_{a \sim \pi_{\theta_t}}(\Delta Q^{\pi_{\theta_t}}(s,g,a))\right)\right.$$

$$\left. - \left(\frac{1}{\alpha}\mathbb{E}_{a \sim \pi_{\theta_t}}[\Delta Q^{\pi_{\theta_t}}(s,g,a)] + \frac{1}{2\alpha^2}\mathrm{Var}_{a \sim \pi_{\theta_t}}(\Delta Q^{\pi_{\theta_t}}(s,g,a))\right)\right]. \tag{32}$$

Simplifying we get:

$$\mathrm{KL}(\pi_{\theta_{t+1}} \parallel \pi_{\theta_t}) \approx \frac{1}{2\alpha^2}\mathbb{E}_{s \sim \mathcal{D}, g \sim \mathcal{G}}\left[\mathrm{Var}_{a \sim \pi_{\theta_t}}(\Delta Q^{\pi_{\theta_t}}(s,g,a))\right]. \tag{33}$$

## B  Task Definition

**FetchReach:** Move the gripper to a target location.

**FetchPickAndPlace:** Pick up a block and place it at a target location.

**FetchPush:** Push the clock to the desired position.

**FetchSlide:** Slide the block to a position outside the robotic arm workspace.

**HandManipulateBlock:** Rotate the block to reach the target rotation in the z-axis.

**HandManipulatePen:** Rotate the pen to reach the target rotation in all axes.

**HandManipulateEgg:** Rotate the egg to reach the target rotation in all axes.

**HandReach:** Move to match a target position for each fingertip.

**Maze:** The environment for navigation tasks is a finite-sized, 2-dimensional maze with blocks. The agent is given a target position and starts from a fixed point in the maze, and it obtains a reward of 0 if it gets sufficiently close to the target position at the current time step or a penalty of -1 otherwise. The agent observes the 2-D coordinates of the maze, and the bounded action space is specified by velocity and direction. The agent moves along the direction with the velocity specified by the action if the new position is not a block and stays otherwise. The maximum time step of an episode is set to 50. MazeA, MazeB, and MazeC variants are shown in Figure 1.

## C  Hyperparameters

Table 2: Hyperparameters for TEACH

| Hyperparameter | Value | Description |
|---|---|---|
| Actor Learning Rate (actor_lr) | $1 \times 10^{-3}$ | Learning rate for the actor-network |
| Critic Learning Rate (critic_lr) | $1 \times 10^{-3}$ | Learning rate for the critic-network |
| Layer Size (hidden) | 512 | Number of neuron in each hidden layer |
| Discount Factor ($\gamma$) | 0.95 | Discount factor for future rewards |
| Replay Buffer Size (buffer_size) | $1 \times 10^{6}$ | Maximum number of transitions in the buffer |
| Target Update Rate ($\tau$) | 0.005 | Update rate for the target networks |
| Noise Scale (action_noise) | 0.2 | Initial scale for Ornstein-Uhlenbeck noise |
| Polyak Averaging Coefficient (polyak) | 0.95 | Interpolation factor for target network updates |
| HER Samples per Transition (future_k) | 4 | Number of hindsight samples per real transition |
| Goal Selection Strategy | Future | Future state is used as the hindsight goal |
| Warmup Steps | 1000 | Number of initial random steps before training |
| Training Frequency (train_freq) | 1 | Train the policy after every step |
| Gradient Clipping | 1.0 | Maximum gradient norm to stabilize training |
| Evaluation Episodes | 20 | Episodes used for evaluation every epoch |

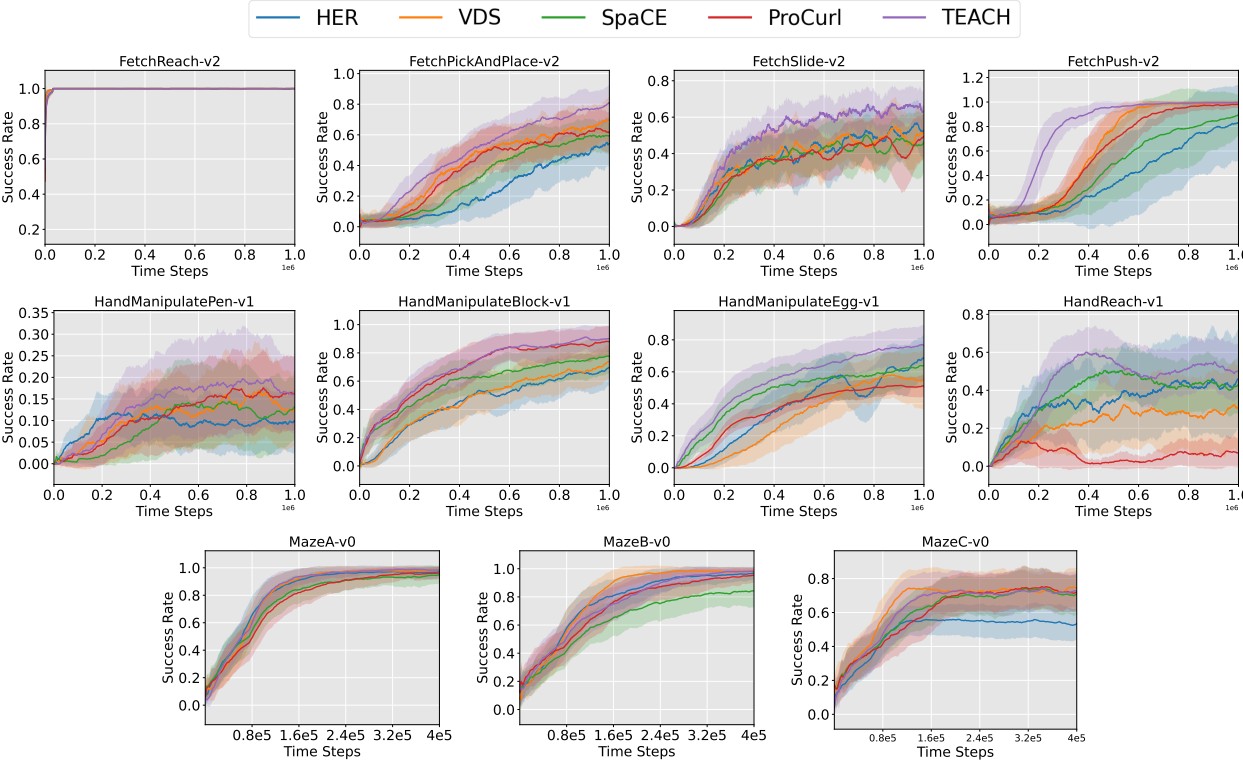

Figure 6: Results show the performance across 8 robotic manipulation tasks and 3 maze navigation tasks. The plots show the success rate along the y-axis evaluated through current policy. The reported results are mean across 5 seeds with shaded regions highlighting standard deviation.

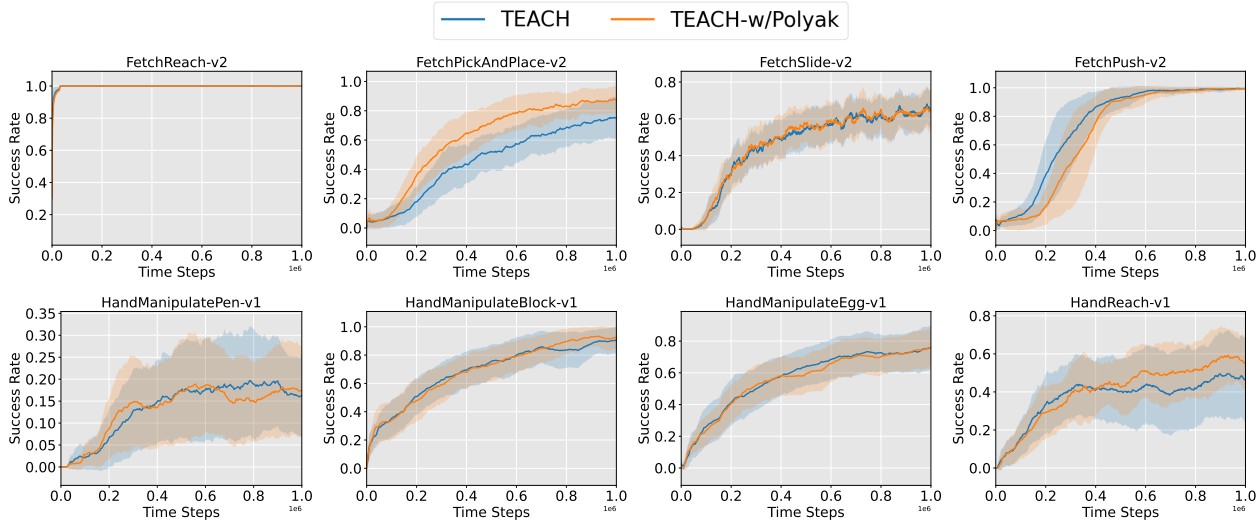

Figure 7: Results show the effect of smooth target confidence score for temporal window size $n = 3$. The results are across 5 seeds where the shaded region represents the standard deviation. We observe that the smooth target confidence score compared to the standard confidence score led to similar performances. This validates the hypothesis that the proposed method measure is robust to noisy value estimates.

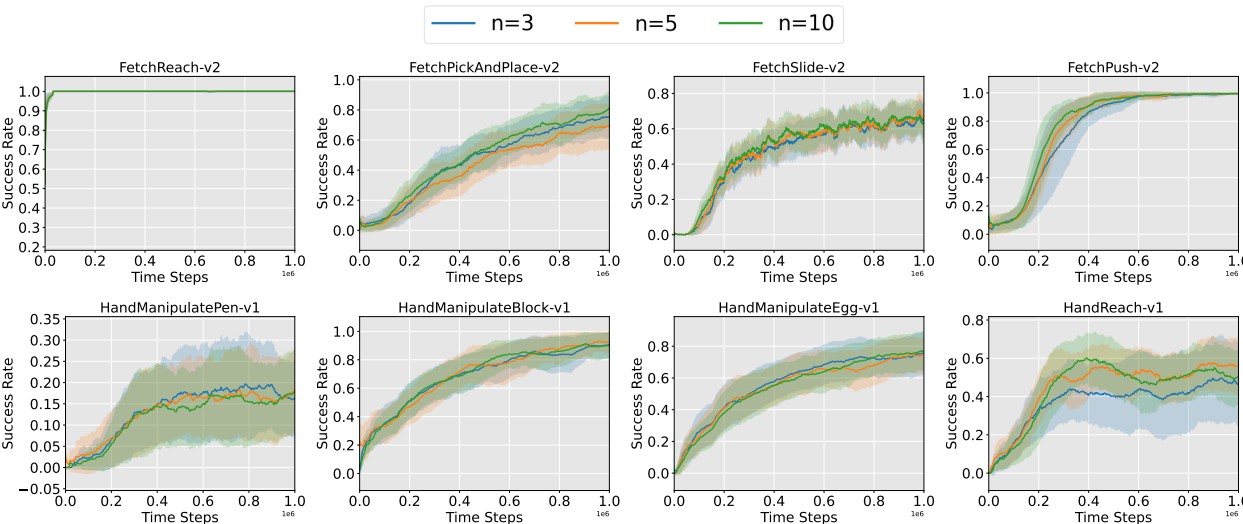

Figure 8: Ablation analysis of the effect of temporal window size $n$. The results are across 5 seeds where the shaded region represents the standard deviation. We observe that the presented approach is insensitive to temporal window size.

