# OpenReview forum: "TEACH: Temporal Variance-Driven Curriculum for Reinforcement Learning"
_TMLR — Rejected by TMLR_

### Review · Reviewer_Ec7P · 2025-02-24

**Summary Of Contributions:**

This paper presents an approach to goal-conditioned policies with curriculum learning. The teacher model dynamically prioritises goals with the highest temporal variance over Q-values. The intuition is that large variance indicates much learning progress is being made. The paper claims convergence guarantees and that the algorithm is model-agnostic. The empirical results show consistent improvements.

I have some reservations about the paper. First, the temporal variance over Q-values does not seem novel, as this seems to be the same as in VDS. Second, the algorithm is not model-agnostic since it relies on a Q-value estimator being available. One may argue that it is not necessary but then one may as well use these Q-values. Third,  the covergence guarantees and proofs in general are questionable.

**Audience:**

Yes

**Broader Impact Concerns:**

There are no broader ethical implications of this work as this is generic machine learning technology.

**Claims And Evidence:**

No

**Requested Changes:**

typos, for example:
varaince --> variance
the the teacher --> the teacher

the rollout trajectory \tau_t = \{(s_t,a_t)\}  --> notation does not make sense since here t refers to the iteration/episode and in other equations the subscript refers to the time step within the episode.

Eq. 7, use of T^\pi(s):  uncommon (and incorrect) to use the same notation as for transition dynamics. The authors have also not defined it.

curriculum learning paragraph: Q(s,g,a) swaps the usual order of Q(s,a,g) (as in your Algorithm 2)

“Since the goal space G is continuous to make the problem intractable, we uniformly sample N goals from the goal
space.”
I think what you mean is that you discretise the goal space so that you have a finite and managable number of goals in the actual goal set to learn.

Theorem 1:
    • this theorem actually does not guarantee convergence for all goals since if training on new goals, the policy will be updated and therefore the actual Q-value of the new policy on previous goals will be changed.
    • in general, the use of the term “time step” seems to indicate the episode (or iteration better). This terminology can be changed as usually time step refers to the time inside the episode.
    • Theorem 1 states “if Var_t(g) = 0” and then proceeds to prove this property. In other words, the proof is trying to prove a precondition.
    •  Var_t, Q_t, and \bar{Q}_t are not clearly defined.
    • The text above Eq. 16 is not grammatically correct and not syntactically connected to Eq. 16
    • It seems strange to talk about “curriculum expansion” only when Var_t \leq \eta when in fact one needs to perform the goal to be able to compute LP. Also why do you distinguish between LP and Var_t? What is the difference?

“except temporal window (n)” --> except the temporal window (n)

the temporal window $n$ and $\Delta t$ do not appear in pseudo-code of algorithm 2 and are not used in any equation. It is therefore difficult to understand how they are being used. The authors mention :
“As shown in Figure 4, the proposed approach demonstrates a high degree of invariance to the size of the temporal window n, which captures the length of past time steps used for computing temporal divergence of policy confidence score”
and
“The interplay frequency controls the temporal interval, after which changes in the policy
confidence score are evaluated.”
I do not understand the difference between these two as it is written now.

Importantly, there seems to be no difference in the technique to the VDS system, other than sort of rearranging the symbols of their approach. VDS uses the standard deviation over time of the Q(s,a,g) for some starting state s and chosen action a. There seems to be no difference at all. Can the authors explain the difference clearly?

At first glance, the approach seems to have strong empirical support. However, there is a lot of ambiguity on what the experiments actually show:
    • hyperparameter settings of the baselines.
    • how do you make sure that the different goals sampled by different methods are the same (at least for the plots/performance comparisons)? If some methods sample different goals at different times, then a lower or higher performance curve would not be surprising since the task may simply be more challenging.


Appendix:

Why does Figure 6 repeat the results of Figure 2?

Eq. 22:
the ratio seems to be incorrect. What is initially a sum over all actions (eq. 21) now has become an expected value.

Eq. 23:
z_t --> Z_t
The authors can mention they expand around x=0 for clarity

Eq. 24 seems to be incorrect. Can the authors please elaborate by writing down all the derivations?

Eq. 25: the actions for t+1 and t are sampled independently, which seems incorrect since in Eq. 20 the actions are sampled from the same policy (which one would expect for a KL divergence metric)

Eq. 26: the authors write an equation of the form x = x + y, which does not make sense

Eq. 27: how is this derived?

**Strengths And Weaknesses:**

Strengths:
- awareness of relevant baselines
- comparatively good empirical performance

Weaknesses:
- the proposed model is not clearly distinguished from VDS
- notations are often confused
- definitions are often missing
- proofs do not seem to be correct
- some experimental details are missing

---

> ### Author Response · Authors · 2025-03-26
> **Response to Reviewer Ec7P**
>
> We sincerely thank the reviewer for their thorough feedback, which has helped us improve the quality of the work. Below, we address each concern and outline the relevant changes made to the revised draft.
>
> **Summary of contribution: Model agnostic and reliance on Q estimator**
>
> We like to clarify that we don't present our method as model-agnostic but as algorithm-agnostic. TEACH is designed to be algorithm-agnostic, relying only on a policy confidence score derived from Q- values estimate—a feature available in most reinforcement learning (RL) frameworks, including Q-learning
> variants (e.g., DQN) and actor-critic methods (e.g., SAC (Haarnoja et al., 2018), PPO (Schulman et al.,
> 2017), TD3 (Fujimoto et al., 2018)). The teacher module uses temporal Q-value variance to guide the
> curriculum, independent of specific algorithmic mechanics like DDPG’s deterministic policy gradients or
> HER’s hindsight relabeling. Our focus on DDPG+HER in the experiments reflects its common use in
> goal-conditioned RL and sparse-reward settings.
>
> **Requested changes and weakness 2: Writing suggestion**
>
> We have corrected all typos (e.g., “varaince” to “variance,” “the the teacher” to “the teacher”) and incorporated notational consistency (e.g., Q(s,a,g), incorrect use of $T^\pi(s)$) to address your writing suggestions. A thorough proofreading ensures no additional errors remain throughout the revised manuscript.
>
> **Requested Changes: clarification regarding Goal space discretization**
>
> We confirm that we discretize the goal space into a finite set of N goals to make the problem tractable, as clarified in Section 4.3.
>
> **Requested changes: what is the difference between the temporal window and interplay play frequency?**
>
> The temporal window (
> n) defines the number of past Q-values used to compute learning progress, while the interplay frequency sets the episode interval for recomputing it. For example, if
> n=10 and the frequency is 5, every 5 episodes, the teacher uses the last 10 Q-values. We’ve updated Algorithm 2 and clarified this in Section 5.6.
>
> **Requested changes and Weakness 1: Difference between VDS and TEACH**
>
> TEACH fundamentally differs from VDS:
>
>  - TEACH leverages the temporal variance of a single Q-function to estimate learning progress for each goal. In contrast, VDS employs an ensemble of Q-functions to compute epistemic uncertainty. It measures the disagreement among these Q-networks.
>
> - TEACH is computationally efficient as it avoids the need for multiple Q-networks, while VDS incurs higher computational costs due to training and maintaining multiple networks.
>
> - TEACH captures learning progress via temporal dynamics, while VDS focuses on epistemic uncertainty, reflecting the agent’s lack of knowledge. This fundamental difference shapes their curriculum generation. TEACH prioritizes goals with high variance, indicating active policy improvement, whereas VDS targets goals where the agent is most unsure, potentially leading to different exploration strategies.
>
> - The theoretical foundation for TEACH, detailed in Section 4.1, links Q-value variance to policy evolution via an approximation of KL divergence, providing a principled basis for its curriculum design. While effective for uncertainty-driven exploration, VDS lacks this temporal dynamic, focusing instead on static ensemble disagreement. This difference is crucial as TEACH’s variance captures ongoing learning, whereas VDS is more aligned with exploration-exploitation trade-offs via uncertainty.
>
> **Requested Changes and weakness 5: Empirical evidence clarification**
>
> The choice of hyperparameters is consistent across methods. The reported results are evaluation curves, not training curves, where goals are sampled uniformly from the goal space. This ensures that reported results remain unbiased to some goal-sampling strategies. Further, the reported results are averaged over 5 different seeds. We use an average of over 20 episodes for each seed at each evaluation step, with each goal sampled uniformly. All these strategies are consistent across all the methods. This ensures that there isn't any bias in evaluating the methods, as detailed in section 5.2.
>
> **Requested changes and weakness 3,4: Correlation Between Q-Value and Policy Divergence (Appendix)**
> - For clarity, we’ve revised the derivation in the appendix, including all steps and assumptions.
>
> - We would like to clarify that Figure 6 does not repeat the results of Figure 2 but provides consolidated results, including those not present in Figure 2.

---

> ### Author Response · Authors · 2025-03-26
>
> **Suggested changes and weakness 3,4: Theorem 1 (revised in Section 4.4)**
>
> **Forgetting:** i.i.d. replay sampling and HER strategy mitigate forgetting by maintaining diverse goal experiences; a discussion on this is added in section 4.4.
>
> **Convergence:** Revised section 4.4 to use standard Q-learning assumptions, ensuring that as the policy improves, variance decreases, guiding curriculum expansion without assuming $\text{Var}^\pi_t(g)$.
>
> **Circular Logic:** The revised proof builds on standard Q learning assumptions. Proof now avoids circularity by relying on policy improvement, which reduces variance naturally.
>
> **Curriculum Expansion:** The "curriculum expansion rule" is a formal construct to prove convergence. It ensures that all goals are eventually included in training, guaranteeing coverage of the entire goal space. In practice, TEACH implicitly expands the curriculum by sampling high LP goals. The $\text{Var}^\pi_t(g)$ is defined when a goal is assumed learned, and LP suggests how to select a new goal once a goal is learned, so they essentially complement each other. These insights are included in the revised section 4.4.
>
>
> [1] Haarnoja, Tuomas, et al. "Soft actor-critic: Off-policy maximum entropy deep reinforcement learning with a stochastic actor." International conference on machine learning. PMLR, 2018.
>
> [2] Schulman, John, et al. "Proximal policy optimization algorithms." arXiv preprint arXiv:1707.06347 (2017).
>
> [3] Fujimoto, Scott, Herke Hoof, and David Meger. "Addressing function approximation error in actor-critic methods." International conference on machine learning. PMLR, 2018.

---

### Review · Reviewer_fqU4 · 2025-03-07

**Summary Of Contributions:**

This paper explores goal-conditioned reinforcement learning (GCRL). The objective is to train a universal goal-conditioned policy that can reach any given goal within a specified horizon, with a binary reward signal provided at the end of each episode. To improve learning efficiency, the authors adopt a teacher-student framework, where the teacher needs to select the next goal in a meaningful order to guide the student's learning process. The main contribution of this work lies in designing a new method for the teacher to perform task selection. In particular, they adopt the learning progress-based idea, which select goals that shows the most progress for the agent.

**Audience:**

Yes

**Broader Impact Concerns:**

Since this is a fundamental RL research study tested on simulated datasets, there are no immediate ethical implications.

**Claims And Evidence:**

No

**Requested Changes:**

Please check the Weakness in the previous section.

**Strengths And Weaknesses:**

## Strength

This work explores an interesting research direction by using automatic curriculum learning (ACL) for goal-conditioned reinforcement learning (GCRL). While this combination has been studied before, the approach proposed in this paper is new.

One major challenge in GCRL is the sparsity of the reward signal since the agent only gets a reward when it successfully reaches the goal within a certain time. The continuous goal space makes the problem even harder than what previous research has considered. To address the problem, the paper suggests a strategy that uses the gradient signal of the policy and the corresponding changes in the Q-value to measure learning progress.

## Weakness

The paper’s formal presentation is weak, making its theoretical contributions unjustified. Key issues include not well-defined notations, mixed theoretical ideas, and an unconvincing convergence analysis.

### Notation Issues:

1. The Q-value notation is inconsistent. if you want to parametrize policy $\pi_{\theta_t}$, the corresponding Q value should be $Q^{\pi_{\theta_t}}$. At the moment, you denote it as $Q^\pi$ and $Q^\pi_t$.

2. The state space is expanded to include goals, but later sections (4.1, 4.2) fail to reflect this in the Q-function.

3. The reward function notation is overloaded that takes different input spaces: $\mathcal{S} \times \mathcal{A}$, $\mathcal{S} \times \mathcal{A} \times \mathcal{S}$, or full trajectories.

4. $\Delta Q$ in the main text is not properly defined.

You might want to reconsider starting from redefining a clearer MDP formulation, such as $M = (\mathcal{S}, \mathcal{A}, \mathcal{G}, r, \gamma, H)$ with $\mathcal{G} \subseteq \mathcal{S}$. To reflect the fact that the reward/return is goal dependent you can define the reward function as $r:\mathcal{S}\times \mathcal{A}\times \mathcal{S}$ such that $r(s,a,g)=1$ for all $s\in\mathcal{S},a\in\mathcal{A}$, and a specified $g$. Otherwise, the agent receive reward $0$. A problem of such modelling, however, can be that it means at each episode, a different goal is given, and hence a different reward function is defined, i.e., the reward function of the MDP is goal-dependent, and it becomes a bit more tricker to talk about the value function and the convergence.

### Ideas Inconsistencies:

1. In section 4.1, you give the expression of the classic “hard” Bellman equation and value, while in section 4.2, you motivate your variance of Q expression using the “soft” update, originated from soft Bellman equation.

2. Section 4.3 restates standard policy gradient results without strong ties to previous sections.

3. In Section 4.4, you suddenly turn to define the learning progress using a goal-dependent Q value but not something related to the “difference” or “variance” function you motivated before.

The inconsistency makes the arguments not well-grounded.

### The convergence analysis seems pointless

First of all, the $\Var_t(g)$ is not properly defined. Furthermore, regardless of the function’s true meaning, the “if” condition in the Theorem is very strong. If an algorithm managed to make $\Var_t(g)$ close to 0 when t is large for all g, then it more or less follows that you achieve a universal goal-conditioned policy, and the statements follow naturally. The real problem is how to achieve a good coverage efficiently for all goals for an algorithm.

---

> ### Author Response · Authors · 2025-03-26
> **Response to Reviewer fqU4**
>
> We appreciate the reviewer’s detailed feedback and have made significant revisions to Section 4 to address the highlighted inconsistencies. Below, we outline how we’ve resolved each concern in the revised draft.
>
> **Notational Inconsistencies**
> To ensure clarity and idea consistency, we’ve updated the notation as follows:
>
> - The Q-function is now consistently denoted as $Q^{\pi_{\theta_t}}$ across sections.
>
> - Goals are explicitly included in the state space and reflected in all Q-function definitions.
>
> - The reward function notation now highlights goal dependency, and $\Delta Q$ is explicitly defined.
>
> - The MDP formulation has been redrafted for a goal-conditioned RL context.
>
> **Idea Inconsistencies**
>
> - Bellman equation Sections 4.1 and 4.2: We use a soft policy update to theoretically analyze the relationship between Q-value variance and policy evolution, as it provides a tractable approximation for policy divergence. In practice, however, our training employs a deterministic policy via DDPG. This distinction is now clearly explained in Section 4.1, aligning the “soft” update’s role in variance analysis with the “hard” Bellman equation in Section 4.2.
>
> - We’ve revised Section 4.3 to define learning progress directly in terms of the temporal variance of goal-dependent Q-values. This ensures consistency with the variance functions introduced earlier and clarifies how the curriculum prioritizes goals where the policy is actively improving.
>
> **convergence analysis:**  We’ve revised the convergence analysis to rely on standard Q-learning assumptions rather than the strong condition $\text{Var}^\pi_t(g)\rightarrow 0$. The updated theorem (Section 4.4) demonstrates that the curriculum efficiently achieves goal coverage by leveraging policy improvements driven by variance, avoiding circularity by not pre-assuming low variance across all goals.
>
> These changes, detailed in the revised draft, resolve the inconsistencies and strengthen the paper’s coherence and theoretical grounding.

---

### Review · Reviewer_kCcJ · 2025-03-21

**Summary Of Contributions:**

The authors introduce TEACH (Temporal Variance-driven Automatic Curriculum Teacher), a novel approach for Goal-Conditioned Reinforcement Learning (GCRL) in multi-goal settings. It focuses on prioritizing goals using high temporal variance in Q-values that improves learning efficiency by targeting high-uncertainty goals. TEACH is a more adaptive and focused learning process that addresses the limitations of noisy value estimates in traditional curriculum learning. Authors also provide a theoretical connection between Q-value changes and policy evolution. Their experiments show that TEACH outperforms existing curriculum learning techniques on different multi-goal sparse reward scenarios.

**Audience:**

Yes

**Claims And Evidence:**

Yes

**Requested Changes:**

- Can you suggest the cases where TEACH may fail?
- Maybe explain the policy confidence score more intuitively.
- Please address the weaknesses/questions from previous section.


There are some minor notations/writing related errors:
- Section 3 Curriculum learning:  “the the”
- 5.1 Space: “curriculum. Which uses”
- Section 4.4 Curriculum Expansion Criterion: “and Once“

**Strengths And Weaknesses:**

Strengths:
- The paper is mostly well-written, with a clear methodology.
- It provides a formal statement and establishes theoretical connections between Q-value temporal variance and policy evolution.
- The empirical results show clear improvements across diverse environments.

Weaknesses / Questions:
- While the method can be integrated into various RL frameworks without major modifications, the experiments mostly use DDPG+HER, but it’s not clear how well TEACH works with other RL algorithms.
- The dimensionality of goal space remains limited to 1e3. How would TEACH perform on high-dimensional goal spaces?
- The method uses Q-value variance to mitigate noise, but cases with highly unstable value estimates might still degrade performance.
- Can you elaborate on why TEACH remains invariant with the size of the temporal history of the value function?
- Maybe I misunderstood something here, but can you explain how the threshold $\eta$ controls the stability of goal learning, and what happens when $Var_t(g) ≤ \eta$ for all goals?
- What could be sources of uncertainty and instability in curriculum learning? Can we draw a direct comparison of different uncertainties? While TEACH is validated in simulated environments, can it be adapted to address real-world uncertainties?
- What additional computational overhead does TEACH introduce compared to others?
- Can TEACH lead to catastrophic forgetting of previously learned goals?
- Based on the analysis, would it be possible to have Adaptive temporal window size (n) selection?
- Does TEACH promote broad exploration or lead to local exploitation of specific goal regions, given its emphasis on high-variance goals?

---

> ### Author Response · Authors · 2025-03-26
> **Response to Reviewer kCcJ**
>
> We appreciate the reviewer's thoughtful questions and suggestions. Below, we address each of the questions and suggestions you provided.
>
> **Weakness/Question 1: Compatibility with RL Algorithms Beyond DDPG+HER**
>
> TEACH is designed to be algorithm-agnostic, relying only on a policy confidence score derived from Q-value estimates—a feature available in most reinforcement learning (RL) frameworks, including Q-learning variants (e.g., DQN) and actor-critic methods (e.g., SAC (Haarnoja et al., 2018), PPO (Schulman et al., 2017), TD3 (Fujimoto et al., 2018)). The teacher module uses temporal Q-value variance to guide the curriculum, independent of specific algorithmic mechanics like DDPG’s deterministic policy gradients or HER’s hindsight relabeling. Our focus on DDPG+HER in the experiments reflects its common use in goal-conditioned RL and sparse-reward settings, where TEACH’s curriculum excels. However, the method’s theoretical foundation supports broader applicability—for example, SAC’s soft Q-values or PPO’s value estimates could similarly inform the curriculum.
>
> **Weakness/Question 2: Performance in High-Dimensional Goal Spaces**
>
> In our experiments, we discretize the continuous goal space $\mathcal{G}$ by sampling $N= 10^3$ goals to ensure computational feasibility. In very high-dimensional spaces (e.g., $10^6$) or continuous spaces without discretization, noise in Q-value estimates might increase, potentially affecting variance-based prioritization. Additionally, keeping a very high dimensionality of goal space will produce too close or overlapping goals. This will essentially cause conflicting Q-values and potentially reflect a non-discriminative behavior of Q value for different goals, which may result in poor performance.
>
> **Weakness/Question 3: Handling Highly Unstable Q-Value Estimates**
>
> TEACH leverages temporal variance over $n$ timesteps to smooth Q-value noise, as demonstrated in our experiments, where it remains robust across noisy and smoothed Q-value conditions. In cases of extreme instability (e.g., divergent training), variance might reflect optimization artifacts rather than policy improvement, potentially impacting performance. To address this, we suggest pairing TEACH with stabilization techniques like TD3’s twin Q-critics or increasing the temporal window $n$ to further dampen fluctuations.
>
> **Weakness/Question 4: Invariance to Temporal Window Size $n$**
>
> Our results show TEACH’s performance is largely insensitive to the temporal window size $n$, with
> $n=10$ slightly optimal. This invariance arises from the convergence properties of Q-values and state-independent estimates of Q-values. Also as the policy improves and $Q^{\pi_{\theta_t}}$ stabilizes the variance plateaus, reducing the sensitivity of the learning progress metric $\text{LP}^\pi(g,t)$ to $n$ beyond a certain point.
>
> **Weakness/Question 5: Role of Threshold $\eta$ in Goal Learning Stability:**
>
> The threshold $\eta$ defines when a goal is “mastered” by ensuring its Q-value variance $\text{Var}^\pi_t(g)$ falls below a level indicating stability, allowing the curriculum to expand to more challenging goals. When $\text{Var}^\pi_t(g) \leq \eta$ for all goals in $\mathcal{G}$, the curriculum effectively encompasses the entire goal space, and training proceeds uniformly across all goals, converging to the optimal policy $\pi^*$. In practice, TEACH dynamically prioritizes high-learning-potential goals without maintaining a fixed curriculum set, avoiding premature saturation.
>
> **Weakness/Question 6: Sources of Uncertainty and Real-World Adaptation**
>
> Key sources of uncertainty in curriculum learning include:
> - Q-value noise: Small replay batches or sparse rewards can introduce fluctuations (mitigated by temporal variance).
>
> - Environment stochasticity: Random dynamics may inflate variance, skewing learning potential (untested in our work).
>
> - Goal sampling bias: Non-uniform sampling could prioritize certain regions (addressed by initial uniform sampling).
>
> Comparing these directly is challenging due to their context-specific impacts. Still, Q-value noise is typically the dominant factor in simulated RL, while stochasticity may dominate in real-world settings. TEACH’s robustness to noise suggests adaptability to real-world uncertainties (e.g., sensor noise, dynamics shifts) when combined with stable RL algorithms or techniques like domain randomization. We will explore real-world validation in future work.
>
> [1] Haarnoja, Tuomas, et al. "Soft actor-critic: Off-policy maximum entropy deep reinforcement learning with a stochastic actor." International conference on machine learning. PMLR, 2018.
>
> [2] Schulman, John, et al. "Proximal policy optimization algorithms." arXiv preprint arXiv:1707.06347 (2017).
>
> [3] Fujimoto, Scott, Herke Hoof, and David Meger. "Addressing function approximation error in actor-critic methods." International conference on machine learning. PMLR, 2018.

---

> > ### Author Response · Authors · 2025-03-26
> >
> > **Weakness/Question 7: Computational Overhead of TEACH**
> >
> > TEACH adds minimal overhead, with variance computation requiring $\mathcal{O}(N\cdot n)$ operations per curriculum update, Which is similar to VDS $\mathcal{O}(N \cdot k)$ with $k$ be the size of Q-ensemble, but they also need to train these additional Q-networks while TEACH does not introduce any new trainable networks. By leveraging a single Q-function’s temporal history, TEACH avoids the complexity of multiple models.
> >
> > **Weakness/Question 8: Risk of Catastrophic Forgetting**
> >
> >  TEACH prevents catastrophic forgetting through (1) HER’s trajectory relabeling, which maintains diverse goal experiences, and (2) uniform replay sampling, ensuring past goals remain in training batches. While TEACH prioritizes high-variance goals, low-variance (mastered) goals are revisited via HER, preserving performance. Empirical results across tasks confirm this stability. We have emphasized these mechanisms in the revised paper (section 4.4) to ensure their effectiveness.
> >
> > **Weakness/Question 9: Adaptive Temporal Window Size**
> >
> > Adaptive $n$ selection is feasible and could enhance TEACH’s flexibility. For instance, increasing $n$ in noisy or difficult environments or decreasing it in rapidly evolving tasks could optimize performance. Variance trends could guide this adaptation—e.g., enlarging $n$ if $\delta_{\mathcal{C}_\pi}$ fluctuates widely.
> >
> > **Weakness/Question 10: Exploration vs. Exploitation**
> >
> > TEACH fosters broad exploration by prioritizing goals with high learning potential, typically at the agent’s skill frontier. HER’s relabeling and uniform initial goal sampling prevents local exploitation by ensuring diverse goal exposure. This balance is reflected in our results, where TEACH outperforms baselines across varied tasks.  The reported results in Figure 2 are test curves where goals are uniformly sampled without curriculum. These findings suggest that TEACH does not introduce any biases in learning.
> >
> > **Requested Changes 1: cases where TEACH may fail?**
> > - In tasks where only a small subset of goals is achievable—like a maze with a single exit—TEACH’s approach of sampling a fixed number of goals (e.g., 1,000) may overrepresent impossible or irrelevant goals. This could cause the curriculum to stall, as it prioritizes unlearnable targets over feasible ones.
> >
> > - For tasks with complex, high-dimensional goals (e.g., image-based goals in robotic tasks), sampling a limited number of goals might fail to adequately cover the goal space. This risks missing critical regions, leading to suboptimal prioritization and incomplete learning.
> >
> > - In environments with local optima—such as a maze with dead ends—high Q-value variance might persist in suboptimal regions. This could trap TEACH’s curriculum on misleading goals, hindering progress toward the global optimum.
> >
> >
> > These limitations arise from TEACH’s design, particularly its fixed goal-sampling strategy and variance-based prioritization, which may not generalize well across all task types. Addressing them could involve adaptive goal sampling or mechanisms to mitigate variance misguidance in complex scenarios.
> >
> > **Requested Changes 2:  explain the policy confidence score more intuitively**  We have revised section 4.2 for better clarity.
> >
> > **Requested Changes 3: notation/writing-related errors** In the revised draft, we have incorporated all the writing and notation-related changes suggested.

---

### Decision · Action_Editor_WXUp · 2025-06-01

**Recommendation:** Reject

**Additional Comments:**

Summary: This paper proposes TEACH (Temporal Variance-aware Curriculum in Offline Reinforcement Learning), a curriculum learning framework for goal-conditioned reinforcement learning (GCRL) in sparse reward environments. TEACH prioritizes training goals based on temporal variance in Q-values, under the hypothesis that high variance corresponds to active learning regions. The method builds a teacher-student loop, where the teacher selects goals with the highest estimated learning progress, and the student updates its policy accordingly. A theoretical link is drawn between Q-value variance and policy evolution, and the authors claim convergence guarantees under certain assumptions. Empirical evaluations on benchmark tasks from the D4RL suite show that TEACH outperforms baseline curriculum learning approaches in terms of sample efficiency and final performance.

Comments: We received three expert reviews for this paper. While reviewers acknowledged the importance of the problem addressed in the paper, they raised substantive concerns regarding the theoretical soundness, novelty, and clarity of the paper. The most consistent criticism centers on the lack of theoretical rigor and inconsistent formalism. The theoretical analysis, particularly Theorem 1, is viewed as weak or vacuous, essentially assuming success rather than providing insight into how TEACH enables convergence. Reviewers noted that key quantities such as the learning progress signal and the variance estimator are insufficiently defined, inconsistently motivated, or disconnected from the overall algorithmic narrative. Moreover, several parts of the theoretical discussion inconsistently invoke deterministic policy gradients, soft policy updates, and Q-learning assumptions without reconciling their differences. A second major concern is the unclear contribution over prior work, particularly Variance-Driven Scheduling (VDS). Empirically, TEACH shows improvement over baselines on selected tasks. However, the evaluation setup lacks important experimental details, such as how goal sets are matched across methods and how hyperparameters are tuned for baselines.  Without this information and broader comparisons, it is difficult to confidently attribute performance gains to the proposed curriculum mechanism. In summary, while the paper explores an important problem and demonstrates promising empirical trends, it suffers from a lack of theoretical clarity, unclear novelty, and insufficient empirical rigor.

**Audience:**

No

**Audience Explanation:**

This paper is likely to interest a subset of TMLR’s audience, particularly researchers working on improving the sample efficiency of RL.  The paper tackles curriculum design in goal-conditioned reinforcement learning (GCRL), a topic of clear interest to researchers working on sample-efficient RL, multi-goal learning, and automatic curriculum learning. The general idea of using learning progress or temporal variance for goal prioritization is interesting. However, the current presentation limits its impact. Due to unclear theoretical grounding, ambiguous notations, and lack of compelling novelty over closely related prior work (e.g., VDS), the findings are unlikely to substantially influence the TMLR readership in their current form.

**Claims And Evidence:**

No

**Claims Explanation:**

The major claims of the papers are the following:

1. The paper claims that the proposed TEACH algorithm and its analysis provide a theoretical foundation linking temporal variance and policy improvement. However, all reviewers have noted that the claimed theoretical contributions are ambiguous and do not have clear evidence. Notations are poorly defined and often inconsistent. Theorem 1 effectively restates that convergence occurs if all goals are learned, without providing actionable insight into how TEACH facilitates that learning more effectively than baseline methods.

2. The paper claims that temporal variance in Q-values provides a reliable signal for goal selection in curriculum learning. While the intuition is reasonable, the paper fails to differentiate this approach clearly from prior work such as VDS, which uses a similar idea. Even though TEACH demonstrates improved performance on selected tasks, it's unclear whether these improvements are due to the curriculum itself or other uncontrolled factors (e.g., goal sampling strategies or hyperparameter tuning). A more thorough comparisonof  baseline algorithms such as VDS is also lacking.

3. The paper proposes a curriculum strategy based on Learning Progress (LP) and claims that it mitigates the impact of noisy value estimates. While the paper introduces a method that uses the temporal variance of Q-values as a proxy for learning progress, the connection between LP and noise mitigation is not clearly established. The paper does not provide either theoretical justification or empirical analysis demonstrating that TEACH reliably reduces the impact of noisy Q-value estimates. More than one reviewer raised concerns that using variance could exacerbate instability in cases with inherently noisy or poorly estimated Q-values. Furthermore, the distinction between variance-driven LP and true learning progress remains conceptually vague.